# Youtu-GraphRAG:
# Vertically Unified Agents for Graph Retrieval-Augmented Complex Reasoning

**Junnan Dong**[1†], **Siyu An**[1†*] **Yifei Yu**[1], **Qian-Wen Zhang**[1], **Linhao Luo**[2],
**Xiao Huang**[3], **Yunsheng Wu**[1], **Di Yin**[1], **Xing Sun**[1]
[1]Tencent Youtu Lab, [2]Monash University, [3]The Hong Kong Polytechnic University
{hansonjdong, siyuan}@tencent.com

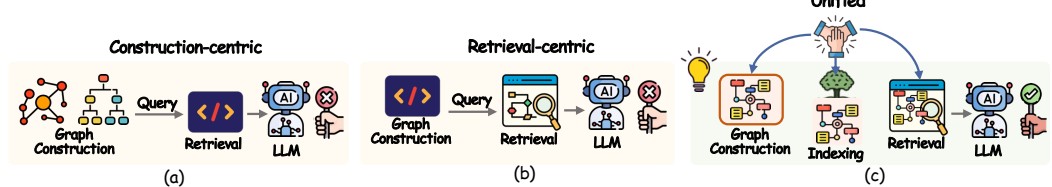

Figure 1: A sketched comparison with existing pipelines. ⟨⟩ represents a non-tailored component, indicating current methods focus on either graph construction (a) or retrieval (b) in isolation, while our proposed `Youtu-GraphRAG` proposes a unified paradigm (c) for superior complex reasoning performance.

## Abstract

Graph retrieval-augmented generation (GraphRAG) has effectively enhanced large language models in complex reasoning by organizing fragmented knowledge into explicitly structured graphs. Prior efforts have been made to improve either graph construction or graph retrieval in isolation, yielding suboptimal performance, especially when domain shifts occur. In this paper, we propose a vertically unified agentic paradigm, `Youtu-GraphRAG`, to jointly connect the entire framework as an intricate integration. Specifically, $(i)$ a seed graph schema is introduced to bound the automatic extraction agent with targeted entity types, relations and attribute types, also continuously expanded for scalability over unseen domains; $(ii)$ To obtain higher-level knowledge upon the schema, we develop novel dually-perceived community detection, fusing structural topology with subgraph semantics for comprehensive knowledge organization. This naturally yields a hierarchical knowledge tree that supports both top-down filtering and bottom-up reasoning with community summaries; $(iii)$ An agentic retriever is designed to interpret the same graph schema to transform complex queries into tractable and parallel sub-queries. It iteratively performs reflection for more advanced reasoning; Extensive experiments across six challenging benchmarks demonstrate the robustness of `Youtu-GraphRAG`, remarkably moving the Pareto frontier of performance and efficiency with up to 33.60% saving of token costs and 16.62% higher accuracy over state-of-the-art baselines. The results indicate our adaptability, allowing seamless domain transfer with minimal intervention on schema.

## 1 Introduction

Graph retrieval-augmented generation (GraphRAG) has emerged as a promising paradigm to enhance large language models (LLMs) with structured knowledge (Xiao et al., 2025; Lu et al., 2026), particularly for complex multi-hop reasoning tasks across multiple documents (Wang et al., 2024; Zhang et al., 2024). By representing fragmented documents as connected graphs with underlying relations (He et al., 2024; Dong et al., 2023), GraphRAG enables LLMs to traverse explicit paths among documents and entities, performing complex reasoning that is otherwise infeasible within flat retrieval (Peng et al., 2024; Han et al., 2024). The structured approach effectively addresses critical

---

*Corresponding author.

limitations in conventional RAG (Gao et al., 2023; Dong et al., 2024c), which often struggles with the coherent relations between discrete pieces of information and multi-hop reasoning.

The evolution of GraphRAG brings two distinct but equally important trajectories since the foundational work of (Edge et al., 2024). First, in terms of retrieval, LightRAG (Guo et al., 2024) pioneered vector sparsification to improve efficiency. While GNN-RAG and GFM-RAG advanced this direction further by incorporating graph neural networks (Mavromatis & Karypis, 2024; Luo et al., 2025) for fine-grained node matching, more recent HippoRAG 1&2 (Jimenez Gutierrez et al., 2024; Gutiérrez et al., 2025) introduced memory and personalized PageRank algorithms for context-aware retrieval. Second, in terms of graph construction, existing methods can be broadly categorized into flat and hierarchical approaches. Early methods, such as KGP (Wang et al., 2024), rely on existing hyperlinks or KNN-based graphs, resulting in coarse-grained relations that fail to capture nuanced hierarchical semantics. More recent advancements, such as GraphRAG (Edge et al., 2024), including community detection and summarization for multi-level information. Followed by hierarchical methods like RAPTOR (Sarthi et al., 2024) and $E^2$GraphRAG (Zhao et al., 2025), they further refine the graph using tree-like clustering and recursive summarization to enrich structural representation. However, both pipelines remain constrained by their isolated optimizations, concentrating on either construction or retrieval while neglecting their interdependencies. This potentially limits complex reasoning performance where the cohesive components are equally important to GraphRAG.

To bridge this gap, we aim to answer a critical question:

***How can we effectively unify graph construction and retrieval for robust complex reasoning?***
This task is challenging for two reasons. First, construction and retrieval are not readily aligned as two distinct components. It remains difficult to organically establish synergy between them, where the constructed graph could effectively benefit retrieval with both structures and semantics. Second, how to properly evaluate the performance remains a tough problem. With the rapid scaling of LLMs, almost all the existing datasets have already been 'seen' before during the pre-training stage of LLMs. This fails to reflect the real performance of the entire GraphRAG.

In this paper, we propose a vertically unified agentic paradigm, `Youtu-GraphRAG`, to jointly consider both graph construction and retrieval as an intricate integration based on graph schema. To be specific, $(i)$ a graph schema is introduced to bound the extraction agent that ensures the quality and conciseness with targeted entity types, relations and attribute types; The seed schema is continuously and automatically expanded based on the feedback. $(ii)$ To obtain higher-level knowledge upon the schema, we develop dually-perceived community detection, fusing structural topology with subgraph semantics for comprehensive knowledge clustering. This naturally yields a hierarchical knowledge tree that supports both top-down filtering and bottom-up reasoning with community summaries; $(iii)$ An agentic retriever is designed to interpret the same graph schema to transform complex queries into parallel sub-queries and perform iterative reflection. The agent iteratively performs both reasoning and reflection for more advanced performance; $(iv)$ To alleviate the knowledge leaking problem in pre-trained LLM, we first propose a tailored anonymous dataset with an 'Anonymity Reversion' task. The model is required to revert the anonymized entities back to its original form with correct, specific named entities. Extensive experiments across six challenging benchmarks demonstrate the robustness of `Youtu-GraphRAG`, remarkably moving the *Pareto frontier* of performance and efficiency with up to 33.60% saving of token consumption and 16.62% higher accuracy over SOTA baselines. The results also indicate our remarkable adaptability which allows seamless domain transfer with minimal intervention on the graph schema, providing insights of the next evolutionary paradigm for real-world applications.

**Contributions**. In general, our primary contributions are summarized hereunder:

- We first propose a vertically unified Agentic GraphRAG framework to integrate graph construction and retrieval for more robust and advanced reasoning, where both construction and retrieval agents are bounded by graph schema for effective extraction and query decomposition, respectively;

- A novel community detection algorithm is employed to inject high-level summarization upon graph schema, simultaneously preserving structural and semantic graph properties;

- We present a tailored anonymous dataset and 'Anonymous Revertion' task is proposed to prevent LLM knowledge leaking for fair evaluation of the GraphRAG performance;

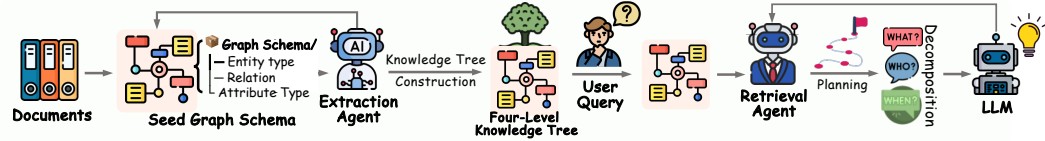

Figure 2: A toy overview of `Youtu-GraphRAG` that unifies graph construction and retrieval through a schema-guided agentic paradigm. $(i)$ An extraction agent automatically processes documents into structured knowledge via targeted entity/relation extraction; $(ii)$ A four-level knowledge tree is constructed upon the schema with a community detection that fuses topological structures and graph semantics, enabling hierarchical reasoning; $(iii)$ A retrieval agent decomposes user queries into parallel sub-queries aligned with the schema, iteratively driving multi-route retrieval.

- Extensive empirical experiments are conducted over five challenging benchmarks, showing state-of-the-art performance across diverse reasoning tasks and domains that moves the Pareto frontier with up to 33.60% saving of token costs and 16.62% higher accuracy.

## 2 TASK DEFINITION

In this section, we formally define the general GraphRAG pipeline with standardized notations from scratch, including both graph construction and graph retrieval. We denote scalars as lowercase alphabets (e.g., $a$), vectors as boldface lowercase alphabets (e.g., $\mathbf{a}$), matrices as boldface uppercase alphabets (e.g., $\mathbf{A}$) and copperplate for a set of elements (e.g., $\mathcal{A}$). We refer to GraphRAG as the task of answering a natural language question by first retrieving structured knowledge from a corpus and then generating a response.

> Given a set of documents $\mathcal{D}$, GraphRAG first leverages a frozen LLM $f_{\text{LLM}}(\cdot)$ to extract important knowledge, connected by a structured graph $\mathcal{G}$ as output. To enrich the understanding of $\mathcal{G}$, a community detection algorithm $f_{\text{comm}}(\mathcal{G})$ is employed to partition $\mathcal{G}$ into communities $\mathcal{C} = \{\mathcal{C}_1, \mathcal{C}_2 \dots \mathcal{C}_m\}$ to obtain higher-level summarizations. Based on the constructed graph $\mathcal{G}$, given a complex query $q \in \mathcal{Q}$, a retrieval model $f_{\text{retrieve}}(q, \mathcal{G}) = \arg\max \mathcal{P}(\mathcal{G}_{\text{sub}} \mid \mathbf{q})$ traverses the graph and retrieves top-$k$ question-specific subgraphs $\mathcal{G}_{sub} \subseteq \mathcal{G}$ that maximize the similarity with given query $q$. The final performance is evaluated from multiple aspects: $(i)$ graph construction costs including time efficiency and token consumptions; $(ii)$ retrieval accuracy and efficiency; and $(iii)$ final answer accuracy comparing $a_{\text{pred}}$ and ground-truths $a_{\text{gold}}$.

### 2.1 CONSTRUCTION STAGE

Contemporary GraphRAG frameworks process document corpora $\mathcal{D}$ through two complementary granularity levels of knowledge organization. At the fine-grained level, a triple-structured graph $\mathcal{G}_{\text{triple}} = (\mathcal{E}, \mathcal{R}, \mathcal{D})$ is extracted using a frozen LLM agent $f_{\text{LLM}}(d)$, which identifies atomic relational triples $(h, r, t)$ from each document $d \in \mathcal{D}$, with entities $\{h, t\} \in \mathcal{E}$ and relations $r \in \mathcal{R}$ explicitly interconnected to capture detailed relational semantics. In parallel, a coarse-grained document graph $\mathcal{G}_{\text{doc}} = (\mathcal{D}, \mathcal{C})$ is constructed by clustering entire documents to preserve broader contextual information. To derive higher-level semantic abstractions, community detection algorithms (e.g., Louvain, Leiden, GMM) partition $\mathcal{G}$ into communities $\mathcal{C} = \{\mathcal{C}_1, \mathcal{C}_2, \dots, \mathcal{C}_m\}$, each of which is summarized into a meta-node $\hat{e}_i = f_{\text{LLM}}(\mathcal{C}_i)$ via LLM-based condensation. The overall construction efficiency is evaluated based on graph build time $t_{\text{construct}}$ and computational token cost $\$$.

### 2.2 RETRIEVAL STAGE

During inference, given a query $q \in \mathcal{Q}$, the typical retrieval model $f_{\text{retrieve}}(q, \mathcal{G}) = \arg\max \mathcal{P}(d \mid \mathbf{q})$ directly returns the top-$k$ similar documents $\hat{\mathcal{D}} = \{d_1, d_2 \cdots d_k\}$ as the final answer, while graph-based methods provide a more explainable subgraph $\hat{\mathcal{G}}$ for multi-hop path traversal, i.e., $f_{\text{retrieve}}(q, \mathcal{G}) = \arg\max \mathcal{P}(\hat{\mathcal{G}} \mid \mathbf{q})$ where $\hat{\mathcal{G}} = \{e_0 \xrightarrow{r_1} e_1 \xrightarrow{r_2} \cdots \xrightarrow{r_k} e_k\} \in \mathcal{G}$. Based on the retrieved subgraph, $f_{\text{LLM}}(q, \hat{\mathcal{G}})$ is employed to generate the final answer. The final performance is evaluated holistically by the retrieval recall comparing $\hat{\mathcal{G}_{\text{doc}}}$ and ground truth documents $\mathcal{A}_{\text{gold}}^{\text{doc}}$ and answer accuracy by comparing between $a_{\text{pred}}$ and $a_{\text{gold}}$.

## 3 APPROACH: YOUTU-GRAPHRAG

In this section, we elaborate on the core methodology of `Youtu-GraphRAG`, designed to answer two fundamental research questions: $(i)$ How to achieve unified optimization of graph construction and retrieval for higher robustness and generalizability? $(ii)$ How could we enable effective reasoning across different knowledge granularities? Correspondingly, our framework integrates three designs in a vertically unified manner based on **graph schema**. First, a graph schema-bounded agent is designed to ensure construction quality while eliminating noise through automatic expansion. Second, beyond schema, we present a dual-perception community detection that jointly analyzes both topological and semantic similarity to create multi-scale knowledge clusters which form a four-level knowledge tree. Finally, an agentic retriever is designed to effectively decompose questions into schema-aligned atomic sub-queries with parallel retrieval routes and iterative reflection.

### 3.1 SCHEMA-BOUNDED AGENTIC EXTRACTION

Existing GraphRAG methods rely on pure LLMs or OpenIE for entity and relation extraction (Jimenez Gutierrez et al., 2024; Gutiérrez et al., 2025; Luo et al., 2025; Edge et al., 2024), often introducing noise and irrelevant information that compromise graph quality. In contrast, we formulate graph extraction as a constrained generation process guided by a compact, domain-specific seed schema, defined as follows:

$$\mathcal{S} \triangleq \langle \mathcal{S}_e, \mathcal{S}_r, \mathcal{S}_{\text{attr}} \rangle, \tag{1}$$

where $\mathcal{S}_e$ indicates the targeted entity types (e.g., `Person`, `Disease`), $\mathcal{S}_r$ guides the extraction with condensed relations (e.g., `treats`, `causes`), and $\mathcal{S}_{attr}$ lists attribute types that could be attached and used to describe any corresponding entities (e.g., `occupation`, `gender`). A frozen LLM-based agent $f_{\text{LLM}}(\mathcal{S}, \mathcal{D})$ is bounded to identify matched information that appear in $\mathcal{S}$, effectively reducing the open-ended search to a structured space defined by the schema $\mathcal{S}$. This confines the model's extraction process to identifying instances of the predefined types in $\mathcal{S}_e$, $\mathcal{S}_r$, and $\mathcal{S}_{attr}$. Formally, for each document $d$, we obtain a set of triples hereunder:

$$\mathcal{T}(d) = \big\{ (h, r, t), (e, r_{\text{attr}}, e_{\text{attr}}) \mid \{f(h), f(t), f(e)\} \in \mathcal{S}_e, \ \{r, r_{\text{attr}}\} \in \mathcal{S}_r, \ e_{\text{attr}} \in \mathcal{S}_{\text{attr}} \big\}. \tag{2}$$

Therefore, in our paper, we define the graph as $\mathcal{G}_{\text{triple}} = (\mathcal{E}, \mathcal{R}, \mathcal{D})$, where the entity set $\mathcal{E} = \{\mathcal{E}_r, \mathcal{E}_{\text{attr}}\}$ includes both named entities and their attributes, and the relation set $\mathcal{R}$ comprises both entity-entity relations and attribute-linking relations (e.g., `has_attribute`). To enhance scalability and adaptability beyond predefined schemas, we introduce an adaptive agent that dynamically refines the initial schema $\mathcal{S}$ through continuous document interaction. The agent proposes schema expansions by identifying relational patterns in each document $d \in \mathcal{D}$ via an update function:

$$\Delta\mathcal{S} = \langle \Delta\mathcal{S}_e, \Delta\mathcal{S}_r, \Delta\mathcal{S}_{\text{attr}} \rangle = \mathbb{I}[f_{\text{LLM}}(d, \mathcal{S}) \odot \mathcal{S}] \geq \mu, \tag{3}$$

where $\mathcal{S}^{(t)}$ represents the schema at iteration $t$, $\mu = 0.9$ serves as a confidence threshold to control the acceptance of new schema elements. $\Delta\mathcal{S}$ contains candidate expansions for entity types, relations, and attributes, respectively. This dynamic adaptation enables the schema to evolve beyond its initial pre-definitions while maintaining controlled growth, as the agent selectively incorporates only high-confidence patterns that demonstrate sufficient frequency and contextual consistency across documents in the new domain. This could effectively balance between strict schema guidance and flexible knowledge acquisition for unseen domains.

### 3.2 UPON SCHEMA: GRAPH INDEXING WITH KNOWLEDGE TREE

The fine-grained raw graphs could quickly become extremely dense and noisy. Typically, a complementary community detection algorithm $f_{\text{comm}}(\mathcal{G})$ is employed to summarize the knowledge so as to reorganize the graph in communities $\mathcal{C} = \{\mathcal{C}_1, \mathcal{C}_2 \ldots \mathcal{C}_m\}$. Contemporary methods apply Louvain, Leiden, Gaussian Mixture Models (GMM) ((Traag et al., 2019; Sarthi et al., 2024)), etc., operates over $\mathcal{G}$ with sufficient summaries and abstracts generated by $f_{\text{LLM}}(d)$. $\mathcal{C}_i \subseteq \mathcal{G}$ is further summarized into a high-level meta-node $\hat{e}_i = f_{\text{LLM}}(\mathcal{C}_i)$ by $f_{\text{LLM}}(\mathcal{C})$ where $\hat{e}_i \in \mathcal{G}$.

However, existing community detection methods, which primarily rely on structural connectivity while neglecting semantic information, often yield suboptimal partitions in knowledge graphs. To address this, we propose a novel dual-perception framework that simultaneously optimizes topological and semantic coherence through a three-stage process, illustrated in Figure 3. The output is

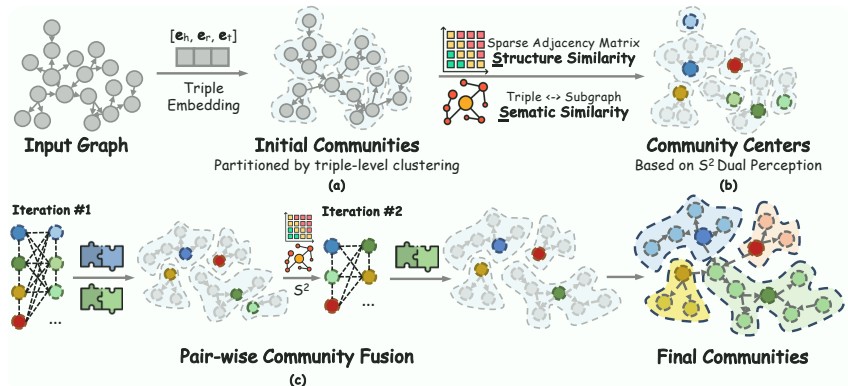

Figure 3: An overview of our dually-perceived community detection. (a) Input graph partitioning into initial communities via triple embeddings; (b) community center identification through joint consideration of topology connectivity and subgraph semantic similarity; and (c) iterative pairwise community merging to form the final hierarchy. Distinct colors represent functionally coherent communities.

compressed into a depth-L *Knowledge Tree* $\mathcal{K}$ ($L = 4$), encapsulating hierarchical levels from coarse community summaries to fine-grained factual leaves, structured as {Community, Keywords, Entity-relation Triples, Attributes}.

**Entity Representation**. Given a graph $\mathcal{G} = (\mathcal{E}, \mathcal{R})$, we first encode each entity $e_i \in \mathcal{E}$ by harvesting its contextualized embedding $\mathbf{e}_i \in \mathbb{R}^d$, aggregating the frozen LM embeddings of all triples within its one-hop neighborhood $\mathcal{N}_i$.

$$\mathbf{e}_i = \frac{1}{|\mathcal{N}_i|} \sum_{(e_i, r, e_j) \in \mathcal{N}_i} f_{\mathrm{LM}}[e_i \| r_{ij} \| e_j]. \tag{4}$$

Specifically, for each triple $(e_i, r, e_j) \in \mathcal{N}_i$, we form a textual sequence and then encode it using a frozen LM $f_{\mathrm{LM}}$, e.g., all-MiniLM-L6-v2 to obtain a contextualized embedding for each triple. To this end, the entity representation could effectively preserve both local structural patterns from one-hop structure and semantic relations via neighbors, enabling clustering based on both signals.

**Cluster Initialization**. To handle large-scale graphs, we first reduce the search space with a initial coarse partition by applying K-means clustering on entity embeddings $\{\mathbf{e}_i\}_{i=1}^{N}$, generating initial communities $\{\mathcal{C}_1^{(0)}, \dots, \mathcal{C}_k^{(0)}\}$. The cluster count $k$ is constrained as $k = \min\left(\max\left(2, \lfloor \mathcal{E}/\beta \rfloor\right), \eta\right)$, with $\beta = 10$ ensuring granularity and $\eta = 200$ preventing over-fragmentation. We use optimized KMeans (n_init=5, random_state=42) for reproducibility.

**Iterative Community Fusion via Dual-Perception Scoring**.

First, to refine the initial clusters, we introduce a dual-perception scoring function $\phi(e_i, \mathcal{C}_m^{(t)})$ that quantifies the affinity between a node $e_i$ and a community $\mathcal{C}_m^{(t)}$ at iteration $t$. This score combines two considerations. $(i)$ **topological connectivity overlap** $(\mathbb{S}_r)$ that measures the Jaccard similarity between the relation incident to $e_i$ and those in $\mathcal{C}_m^{(t)}$; $(ii)$ **subgraph semantic similarity** $(\mathbb{S}_s)$, which computes the cosine similarity between the entity embedding $\mathcal{F}_\Theta(\mathbf{T}_i)$ and the community centroid $\mathbb{E}_{\mathcal{C}_m^{(t)}}[\mathcal{F}_\Theta(\mathbf{T}_{jk})]$, where $\mathcal{F}_\Theta$ is a matrix for embedding transformation.

$$\phi(e_i, \mathcal{C}_m) = \underbrace{\mathbb{S}_r(e_i, \mathcal{C}_m)}_{\text{relational}} \oplus \lambda \underbrace{\mathbb{S}_s(e_i, \mathcal{C}_m)}_{\text{semantic}}, \tag{5}$$

with

$$\begin{aligned} \mathbb{S}_r(e_i, \mathcal{C}_m) &= \frac{\|\Psi(e_i) \cap \Psi(\mathcal{C}_m)\|_2}{\|\Psi(e_i) \cup \Psi(\mathcal{C}_m)\|_2}, \\ \mathbb{S}_s(e_i, \mathcal{C}_m) &= \phi\Big(\mathcal{F}_\Theta(\mathbf{T}_i), \sum_{j \in \mathcal{C}_m} \big(\mathcal{F}_\Theta(\mathbf{T}_j)\big)\Big), \end{aligned} \tag{6}$$

where $\mathbb{S}_s$ denotes the Jaccard similarity matrix computed over the multiset of incident relation types $\Psi(\cdot)$. $\mathbb{S}_s(i, j)$ measures the overlap of relation-specific neighborhoods between nodes $i$ and $j$.

At each iteration $t$, we select the most representative entity $e_{\text{center}}^* = \arg\max \phi(e_i, \mathcal{C}_m)$ for each community based on the dual-perception score, which combines both structural connectivity and

semantic representativeness. Communities are merged if the divergence between their centroid expectation values, formulated hereunder:

$$\mathbb{E}[\phi(e_i, \mathcal{C}_a^{(t)})] - \mathbb{E}[\phi(e_i, \mathcal{C}_b^{(t)})] < \epsilon. \qquad (7)$$

where the merge will only occur when the value is below a threshold $\epsilon$, thereby reducing the matching search space from node-community comparison to node-node comparison, yielding a boosted, efficient hierarchical community detection.

### 3.2.1 KNOWLEDGE TREE

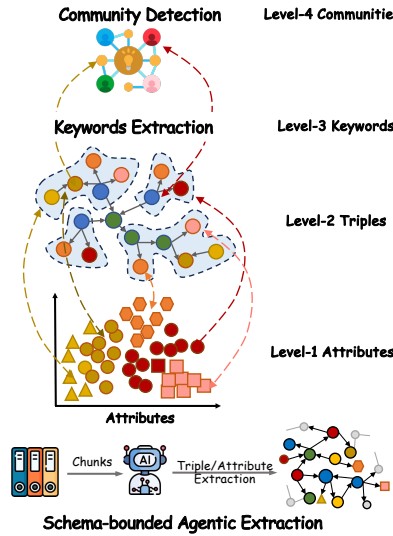

Figure 4: Four-level knowledge tree.

To this end, building upon our schema-bounded extraction framework, we develop a hierarchical knowledge organization pipeline that transforms raw graphs into a structured *Knowledge Tree* $\mathcal{K}$ shown in Figure 4. First, the process begins with our novel dual-perception community detection algorithm, which computes entity-community affinity through the combined metric, blending topological connectivity overlap with semantic subgraph similarity. Second, $f_{\text{LLM}}(\mathcal{C}_m)$ is then applied to generate a brief name and description for the entire community based on the member names. These community names are treated as community nodes and inserted into the original graph, connecting with each member entity with the relation member_of. Third, within each detected community $\mathcal{C}_m$, we identify pivotal keywords by selecting entities maximizing the structural-semantic score $\arg\max_{e_i \in \mathcal{C}_m} \phi(e_i, \mathcal{C}_m)$.

The resulting hierarchy, together with the schema, collectively informs the construction of our four-layer knowledge tree $\mathcal{K}$. The tree maximizes bottom-up semantic coherence at each level, simultaneously preserving fine-grained reasoning through granular entity-relation/entity-attribute retrieval ($\mathcal{L}_1$) and enhancing high-level community-based filtering ($\mathcal{L}_4$). We formally define it as $\mathcal{K} = \bigcup_{\ell=1}^{4} L_\ell$,

$$L_\ell = \begin{cases} \{\mathcal{C}_m\} & \ell = 4 \ (\texttt{Community}) \\ \{\arg\max \phi(v_i, \mathcal{C}_m)\} & \ell = 3 \ (\texttt{Keywords}) \\ \{(h, r, t) \mid h, t \in \mathcal{E}, r \in \mathcal{R}\} & \ell = 2 \ (\texttt{Entity-Relation Triples}) \\ \{(e, \texttt{has\_attr}, \{e_{\text{attr}}^{\text{type}} : e_{\text{attr}}^{\text{value}}\})\} & \ell = 1 \ (\texttt{Attributes}) \end{cases} \qquad (8)$$

### 3.3 AGENTIC RETRIEVER

**Schema-enhanced Query Decomposer**. The complexity of multi-hop queries in large-scale knowledge tree necessitates an intelligent decomposition mechanism that respects both the explicit schema constraints and implicit semantic relationships. We propose a schema-guided decomposition approach illustrated in Appendix A.2. By leveraging the graph schema $\mathcal{S} = (\mathcal{S}_e, \mathcal{S}_r, \mathcal{S}_{\text{attr}})$, we ensure that each generated atomic sub-query strictly adheres to valid patterns in the knowledge tree, filtered by the identified schemas with entity types and attribute types. This schema-awareness prevents the generation of ill-formed queries that would either fail to return results or retrieve irrelevant information. Therefore, the final $\mathcal{Q} = f_{\text{LLM}}(q, \mathcal{S}) = \{q_1, q_2 \dots q_i\}$, where $i$ is a pre-defined maximum number for total atomic sub-queries and each $q_i$ explicitly targets either: $(i)$ node-level retrieval $(e, \texttt{has\_attr}, a)$, $(ii)$ triple-level matching $(h, r, t)$, or $(iii)$ community-level verification $\mathcal{C}_m$, as determined by schema elements $\mathcal{S}_e, \mathcal{S}_r$, and $\mathcal{S}_{attr}$.

**Iterative Reasoning and Reflection**. Since reasoning and reflection are two core cognitive capabilities for the agent, following the standard agent framework of perception-reasoning-action cycles, we formalize our agent as a tuple $\mathcal{A} = \langle \mathcal{H}, f_{\text{LLM}} \rangle$, where $\mathcal{H}$ denotes the agent's historical memory containing both former reasoning step and the retrieval results to derive insights for new actions, and the functions $f_{\text{LLM}}$ is employed to implement both key operations.

$$\mathcal{A}^{(t)} = \underbrace{f_{\text{LLM}}}_{\text{Reasoning}} \big( q^t, \underbrace{\mathcal{H}^{(t-1)}}_{\text{Reflection}} \big), \qquad (9)$$

This process addresses the compositional generalization challenge in complex QA by $(i)$ maintaining explicit symbolic grounding through $\mathcal{S}$ during reasoning steps, and $(ii)$ performing continuous self-monitoring via reflection to detect and correct reasoning paths. The agent's operational flow alternates between forward reasoning with schema-guided query decomposition and retrieval and backward reflection for complex scenarios, creating a closed-loop framework that progressively converges to optimal solutions. To maximize the strength of different granularities, we equip the retrieval with multiple routes, including entity retrieval, triple matching, community filtering and DFS path traversal. The details could be found in the Appendix A.3.

## 4 EXPERIMENTS

To systematically evaluate the model's retrieval and generation capabilities across diverse scenarios, we designed a unified experimental framework encompassing evaluation metrics, datasets, and baseline models. Specifically, we adopt a dual-mode evaluation protocol with both open mode (allowing the use of parametric knowledge) and reject mode (requiring responses strictly based on retrieved evidence) to assess the reliability of GraphRAG methods. Experiments are conducted on multiple established and newly constructed anonymous datasets, with comprehensive comparisons across representative baseline methods. Detailed experimental settings are provided in Appendix A.4.

### 4.0.1 EVALUATION METRICS

Following the workflow of RAG, the evaluation is typically divided into two stages: $(i)$ assess the accuracy of retrieved evidence and $(ii)$ examine the end-to-end performance by evaluating the quality of LLMs responses generated from the retrieved evidence. In practical deployment scenarios, where multiple valid retrieval references may exist for identical answers, the latter evaluation paradigm has emerged as the prevailing standard in practical applications. Regarding the assessment of LLMs responses, several character-based matching protocols, e.g., recall, EM and F1 score were established. To account for semantic deviations caused by minor character variations, where slight textual differences may lead to substantially divergent meanings, we employ DeepSeek-V3-0324 to assess response similarity against ground truth references.

During the reproduction of various GraphRAG frameworks, we observed experimental results exhibit significant variations depending on the prompts in the LLMs generation stage. Specifically, some frameworks(Zhao et al. (2025)) instruct to explicitly reject to answer when retrieved evidence is insufficient, while others(Xiao et al. (2025); Sarthi et al. (2024)) allow LLMs to leverage its parametric knowledge or ambiguates the instruction in such cases. Given that most LLMs have been exposed to extensive corpora during pretraining, we identify answering questions based on LLMs' knowledge rather than retrieval mechanism as a critical factor for fairly evaluation, which we term **knowledge leaking**. To separately assess two critical capabilities: $(i)$ recognizing knowledge limitations, and $(ii)$ leveraging LLMs' parametric knowledge, we therefore implement a dual-mode evaluation on three widely-used datasets:

- **Reject mode**. Under this mode, LLMs must reject to answer the question when retrieval fails to provide sufficient evidence from the given graph. This strictly evaluates the retrieval effectiveness and prevent hallucination among existing models.

- **Open mode**. LLMs are allowed to answer using either retrieved content or its inherently parametric knowledge. This maximally measures the overall capability in real-world practical deployment.

We have reproduced representative baselines and conducted comprehensive evaluations based on the metrics in this work. The corresponding prompts are provided in subsection A.6. Moreover, the observations further underscore the importance of our proposed AnonyRAG dataset to ensure fair and comprehensive assessment of GraphRAG methods.

### 4.1 COMPARISON OF TIME AND TOKEN CONSUMPTION

For baselines involving graph construction and community detection stages, this section compares their token and time consumption. All procedures are executed using 32-thread concurrent inference to ensure both the efficiency of graph construction and the fairness of comparisons. Figure 5a presents the time and token consumption during the graph construction stage for

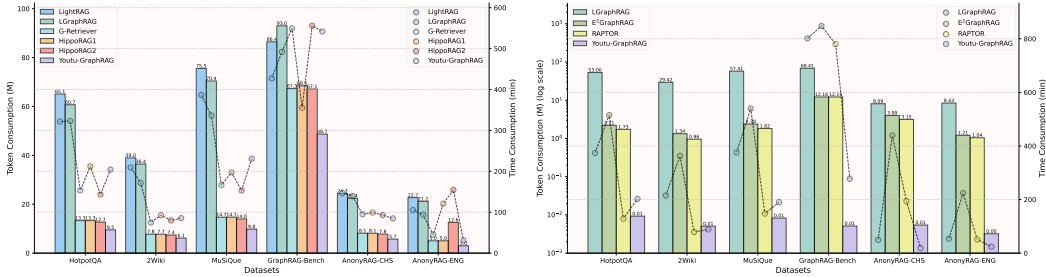

(a) Consumption comparison of graph construction.    (b) Consumption comparison of community detection

`Youtu-GraphRAG` and five baselines. Our method consistently achieves the lowest token consumption across all six datasets and maintains relatively efficient time performance on five of the datasets. In the community detection stage, as shown in Figure 5b, we achieve the lowest token consumption compared with the other three baselines, consuming no more than 10,000 tokens on any dataset. Meanwhile, our method demonstrates consistently efficient performance across all datasets.

Table 1: Overall performance comparisons over benchmark datasets in terms of top-20 Accuracy.

| Method | HotpotQA | | 2Wiki | | MuSiQue | | G-Bench | Annoy-CHS | Annoy-ENG |
|---|---|---|---|---|---|---|---|---|---|
| | Open | Reject | Open | Reject | Open | Reject | Open | Open | Open |
| **Deepseek-V3-0324** | | | | | | | | | |
| Zero-shot LLM | 53.70 | - | 41.6 | - | 25.7 | - | 70.92 | 9.62 | 8.18 |
| Naive RAG | 79.90 | 72.40 | 70.3 | 38.9 | 47.49 | 30.63 | 71.81 | 12.5 | 43.02 |
| E$^2$GraphRAG | 68.70 | 48.80 | 43.20 | 20.00 | 28.36 | 8.01 | 68.66 | 16.01 | 35.97 |
| RAPTOR | 80.90 | 73.60 | 70.10 | 38.40 | 48.50 | 31.10 | 73.08 | 12.08 | 40.2 |
| LightRAG | 71.90 | 56.00 | 58.00 | 29.20 | 38.98 | 24.57 | 70.83 | 9.16 | 22.14 |
| GraphRAG | 56.10 | 26.40 | 41.80 | 10.00 | 32.20 | 16.50 | 75.54 | 21.66 | 38.85 |
| G-Retriever | 49.00 | 6.70 | 35.80 | 5.00 | 23.50 | 1.70 | 70.63 | 4.07 | 5.08 |
| HippoRAG | 81.70 | 73.10 | 77.90 | 64.00 | 48.30 | 36.20 | 72.89 | 36.77 | 40.68 |
| HippoRAG-IRCOT | 81.00 | 74.60 | 78.40 | 66.00 | 46.50 | 35.50 | 73.38 | 36.05 | 42.17 |
| HippoRAG2 | 81.80 | 74.90 | 77.30 | 48.30 | 50.80 | 37.80 | 79.37 | 12.92 | 43.16 |
| Ours w/o Agent | 83.70 | 75.30 | 72.80 | 57.80 | 51.40 | 40.00 | 81.53 | 37.06 | 40.05 |
| Youtu-GraphRAG | 86.50 | 81.20 | 85.50 | 77.60 | 53.60 | 47.50 | 86.54 | 42.88 | 43.26 |
| **Qwen3-32B** | | | | | | | | | |
| Zero-shot LLM | 36.40 | - | 33.30 | - | 13.40 | - | 70.04 | 5.11 | 6.49 |
| Naive RAG | 75.00 | 69.00 | 58.50 | 39.60 | 40.64 | 33.03 | 72.69 | 7.56 | 26.84 |
| RAPTOR | 79.20 | 72.90 | 61.20 | 40.10 | 38.99 | 32.86 | 72.20 | 13.37 | 22.14 |
| HippoRAG | 77.00 | 71.80 | 72.80 | 62.50 | 40.60 | 32.10 | 75.64 | 8.58 | 32.30 |
| HippoRAG-IRCOT | 80.30 | 76.60 | 74.80 | 65.40 | 44.70 | 37.40 | 77.11 | 9. 16 | 33.15 |
| HippoRAG2 | 81.80 | 71.30 | 65.20 | 39.90 | 51.40 | 37.70 | 80.35 | 12.65 | 38.36 |
| Ours w/o Agent | 83.80 | 73.90 | 74.90 | 55.30 | 52.90 | 40.10 | 80.74 | 34.88 | 35.13 |
| Youtu-GraphRAG | 85.90 | 78.60 | 85.70 | 74.20 | 54.60 | 45.30 | 84.48 | 39.24 | 40.05 |

## 4.2 MAIN PERFORMANCE COMPARISON

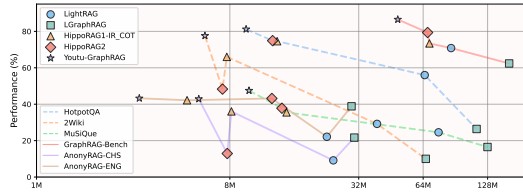

Figure 6: `Youtu-GraphRAG` effectively moves the Pareto frontier with lower token costs and higher performance over six benchmarks.

In Table 1, we report the top-20 accuracy across six challenging benchmarks under both open and reject modes, based on two strong LLM backbones, i.e., DeepSeek-V3-0324 and Qwen3-32b. Across virtually all datasets and settings, `Youtu-GraphRAG` attains the highest performance, reflecting its ability to combine precise retrieval with robust reasoning. Besides, we also include an variant with no agent for iterative reasoning and reflection as a lightweight version, i.e., `Ours w/o Agent`, fulfilling real-world applications requiring real-time interactive feedback. The distinction between the two evaluation modes provides complementary perspectives on system capability. **Open mode** unlocks the full reasoning potential of the LLM to synthesize an answer regardless of retrieval gaps.

This mirrors high-coverage real-world deployments where maximizing end-task accuracy outweighs caution. `Youtu-GraphRAG` consistently outperforms existing baselines, achieving improvements from 2 to 8 points over the strongest competitor across datasets. When augmented with our agent framework, `Youtu-GraphRAG` further pushes the performance frontier, reaching top-20 accuracies of 86.5%, 85.5%, and 53.6% on HotpotQA, 2Wiki, and MuSiQue respectively under Deepseek-V3-0324, and 85.9%, 85.7%, and 54.6% under Qwen3-32B, demonstrating a clear advantage in multi-hop reasoning and cross-document synthesis. **Reject mode**, by contrast, imposes a stringent criterion if the retrieved context is insufficient, the model must abstain. `Youtu-GraphRAG` attains 81.2%, 77.6%, and 47.5% on HotpotQA, 2Wiki, and MuSiQue, outperforming the strongest baseline by 7–14 points. Across all datasets, our method achieves consistently higher top-20 accuracy, confirming its ability to synergize graph-based retrieval with agent-driven reasoning for both high-coverage and high-precision scenarios. We value this metric since it directly probes retrieval quality, as speculative answers are penalized and the acceptance rate becomes a direct function of retrieval completeness and precision. Our superiority on two anonymous datasets also validates the generalizability of `Youtu-GraphRAG` beyond standard benchmarks. Specifically, under the open mode, it achieves 42.88% and 43.26% top-20 accuracy on Annoy-CHS and Annoy-ENG, respectively, surpassing all baselines by a clear margin. These results also reflect our robust reasoning and retrieval integration across diverse languages and domains, demonstrating that our approach could be easily transferred to previously unseen data distributions while maintaining high accuracy.

A key objective of `Youtu-GraphRAG` is to jointly optimize performance and efficiency by unifying graph construction and retrieval. Figure 6 illustrates the trade-off between token consumption during the construction and overall QA performance across six benchmarks. Our approach consistently achieves optimal performance with the least token consumption, effectively shifting the *Pareto frontier* of both QA performance and costs compared to all baselines.

While existing GraphRAG methods struggle to balance graph construction cost and answer accuracy, `Youtu-GraphRAG` introduces a vertically integrated way combining schema-guided extraction, dual-perception community detection, and schema-enhanced agentic retrieval to build and reason over concise yet informative graphs. Our approach shifts the *Pareto frontier*, achieving state-of-the-art performance across all benchmarks while reducing graph construction token usage. These results affirm that synergistic integration of schema alignment, hierarchical knowledge trees, and adaptive retrieval significantly enhances the practicality and efficiency of GraphRAG.

Table 2: Overall performance comparisons based on DeepSeek in terms of top-10 accuracy.

| Method | HotpotQA | | 2Wiki | | MuSiQue | | G-Bench | Annoy-CHS | Annoy-ENG |
|---|---|---|---|---|---|---|---|---|---|
| | Open | Reject | Open | Reject | Open | Reject | Open | Open | Open |
| Naive RAG | 79.40 | 68.00 | 67.60 | 33.70 | 45.58 | 26.73 | 71.22 | 12.08 | 38.93 |
| RAPTOR | 78.20 | 67.10 | 67.40 | 36.40 | 45.88 | 30.03 | 72.79 | 11.77 | 33.99 |
| G-Retriever | 49.90 | 5.90 | 38.00 | 3.80 | 23.50 | 1.70 | 70.24 | 5.38 | 5.50 |
| LightRAG | 71.98 | 58.10 | 65.70 | 38.10 | 39.40 | 22.90 | 69.74 | 8.58 | 18.90 |
| GraphRAG | 54.30 | 23.70 | 40.00 | 9.80 | 30.20 | 16.00 | 61.39 | 21.37 | 38.36 |
| HippoRAG | 78.20 | 69.40 | 77.10 | 61.10 | 45.20 | 30.90 | 70.14 | 34.01 | 40.12 |
| HippoRAG-IRCOT | 78.10 | 70.20 | 77.70 | 60.70 | 44.40 | 31.60 | 72.89 | 36.19 | 41.42 |
| HippoRAG2 | 79.40 | 70.40 | 74.60 | 45.80 | 49.10 | 34.00 | 77.21 | 13.52 | 37.24 |
| Ours w/o Agent | **80.50** | **72.10** | **72.10** | **54.40** | **49.80** | **38.30** | **80.55** | 35.17 | **40.54** |
| Youtu-GraphRAG | **83.40** | **78.90** | **82.30** | **72.60** | **52.10** | **46.90** | **83.50** | 38.08 | **42.57** |

## 4.3 ANALYSIS OF GENERALIZABILITY

To examine the domain-transfer capability of `Youtu-GraphRAG`, we evaluate it across six heterogeneous benchmarks without any task-specific fine-tuning. As shown in Figure 7, `Youtu-GraphRAG` achieves the best performance in both Open Accuracy and Reject Accuracy on all datasets, surpassing state-of-the-art GraphRAG baselines by a clear margin.

We attribute this strong generalizability to the intrinsic integration of graph construction and retrieval: (*i*) schema-guided extraction yields consistent, domain-adaptive graphs, while dual-perception community detection builds robust hierarchical structures; (*ii*) the agentic query decomposer dynamically adapts to various question types. These results confirm that `Youtu-GraphRAG` transfers seamlessly to unseen domains, preserving structural and reasoning fidelity.

Table 3: Ablation studies of our method over six datasets. We evaluate three variants: w/o Community detection, w/o Agent coordination, and w/o Schema guidance.

| Variants | HotptQA | 2Wiki | MuSiQue | G-Bench | AnonyRAG-CHS | AnonyRAG-ENG |
|---|---|---|---|---|---|---|
| w/o Comm. | 79.50 | 75.10 | 44.00 | 85.02 | 39.97 | 39.92 |
| w/o Agent | 75.30 | 57.80 | 40.00 | 81.53 | 37.60 | 40.05 |
| w/o Schema | 77.10 | 73.40 | 45.60 | 83.50 | 35.61 | 40.32 |
| Youtu-GraphRAG | **81.20** | **77.60** | **47.50** | **86.54** | **42.88** | **43.26** |

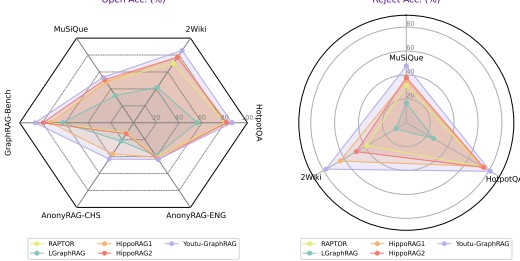

Figure 7: We showcase the generalizability over six benchmarks in terms of both open and reject accuracy.

Furthermore, as shown in Table 2, our method consistently outperforms all baselines in both open and reject modes under top-10 retrieval evaluation. In open mode, Youtu-GraphRAG achieves accuracies of 83.4%, 82.3%, and 52.1% on HotpotQA, 2Wiki, and MuSiQue, outperforming the strongest competitor by 4~8 points. Under reject mode, improvements reach 8~12 points, reflecting stronger retrieval fidelity and reduced speculation. Consistent superiority is also observed on anonymous datasets, e.g., 38.08% on Annoy-CHS, and 42.57% on Annoy-ENG. These results underscore the robustness and adaptability of our proposed schema-based integration of all components, even under stricter evaluation with limited hints.

## 4.4 ABLATION STUDIES

To quantify the contribution of each component, we perform ablations by removing community detection (w/o Comm.), agent reasoning and reflection (w/o Agent), and schema guidance (w/o Schema). Results on six benchmarks are summarized in Table 3.

Specifically, removing community detection leads to a consistent drop across all datasets, particularly on multi-hop QA tasks such as HotpotQA and 2Wiki around 1.7% and 2.5%, indicating that structuring knowledge into coherent communities facilitates more accurate retrieval and reasoning for global questions. The absence of agentic schema expandion, reasoning and reflection causes the most severe degradation on complex reasoning datasets, especially on 2Wiki and MuSiQue with remarkable 19.8% and 7.5% differences, supporting our motivation that the iterative reasoning-feedback loop plays an essential role for resolving ambiguous intermediate steps. Eliminating schema guidance results in noticeable performance drops on knowledge-intensive settings, especially on AnonyRAG-CHS with 7.27% decreases, highlighting the importance of a high-quality initialization of seed schema for new domains. This further demonstrates our advantage that only requires minimum intervention for domain shifts. In conclusion, our model consistently outperforms all ablated variants, demonstrating the importance of each component for multi-hop inference.

## 5 CONCLUSIONS

In this paper, we propose Youtu-GraphRAG, a vertically unified agentic paradigm that jointly optimizes both aspects through a graph schema. Our framework introduces $(i)$ a schema-guided agent for continuous knowledge extraction with predefined entity types, relations, and attributes; $(ii)$ dually-perceived community and keyword detection, fusing structural topology with subgraph semantics to construct a hierarchical knowledge tree that supports top-down filtering and bottom-up reasoning; $(iii)$ an agentic retriever interprets the schema to break complex queries into tractable sub-queries, paired with an iterative reasoning and reflection; and $(iv)$ Anonymity Reversion, a novel task to mitigate knowledge leakage in LLMs, deeply measuring the real performance of GraphRAG frameworks supported by a carefully curated anonymous dataset. Extensive experiments across six challenging benchmarks demonstrate Youtu-GraphRAG's robustness, advancing the Pareto frontier with up to 33.60% reduction in token costs and 16.62% higher accuracy than state-of-the-art baselines. Notably, our framework exhibits strong adaptability, enabling seamless domain transfer with minimal schema adjustments. These results underscore the importance of unified graph construction and retrieval, paving the way for more efficient and generalizable GraphRAG.

ETHICS STATEMENT

Our research addresses technical problems in the field of information retrieval and knowledge representation. The work is mainly about the automated processing of publicly available data and questions, and involves no human subjects, animals, or environmentally sensitive materials. We therefore anticipate no direct physical or environmental ethical risks.

REPRODUCIBILITY STATEMENT

Detailed method design, implementation and reference, including dataset curation and sources, baselines, experimental settings of this paper, are provided in experimental settings in Section 3 and A.4. Baselines have been carefully implemented to ensure fair comparison. The code and data are open-sourced and could be accessed via the anonymous link:
https://github.com/TencentCloudADP/Youtu-GraphRAG

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

# A APPENDIX

## A.1 USAGE OF LLMS

This paper was completely written by human beings and has been proofread and slightly polished with the assistance of LLMs. All LLM-generated content has been thoroughly examined and fact-checked to uphold the accuracy and integrity of the work. We assume full responsibility for any errors or inaccuracies that may remain in the final version.

## A.2 ILLUSTRATION OF THE DECOMPOSER

Consider the query "Where did Turing Award winners study?" Our method automatically maps "Turing Award winner" to the appropriate entity type $\mathcal{S}_e$ : Person with the specific award attribute, while correctly interpreting "study" as an $\mathcal{S}_r$ : educated_at. This semantic precision prevents the common problem of interpretation drift that often occurs in naive decomposition approaches.

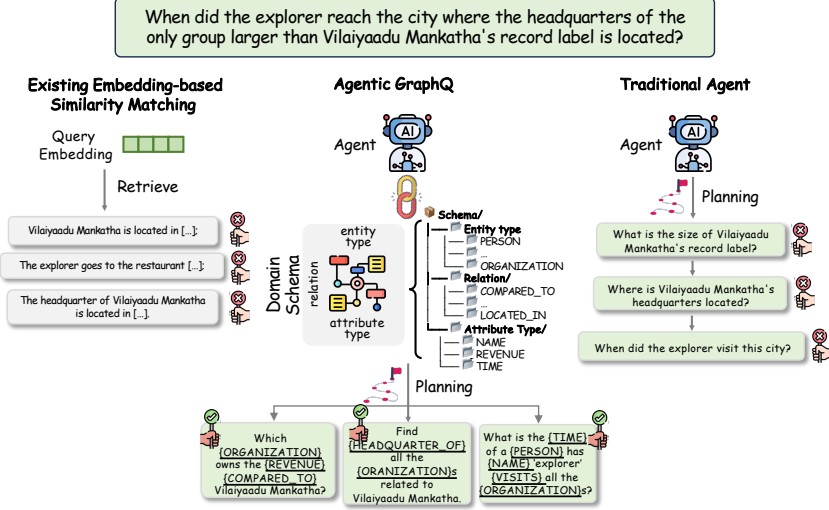

Figure 8: The figure contrasts three query-resolution strategies for a multi-hop question. While embedding matching retrieves disjointed facts (left) and traditional agents use repetitive templates (right), our agentic decomposer (center) leverages domain schema to plan efficient sub-queries: (1) compare record label revenues, (2) locate the larger group's headquarters, and (3) trace the explorer's visit—achieving precise, with parallel reasoning and outperforming unstructured retrieval and template-based agents.

## A.3 MULTI-ROUTE RETRIEVAL

To handle diverse sub-query types, we implement four parallel retrieval strategies with distinct optimization objectives:

$$
\begin{aligned}
\textbf{Entity Matching} &: \quad \arg\max_{e \in \mathcal{E}} \cos(\mathbf{e}, \mathbf{q}_i) \\
\textbf{Triple Matching} &: \quad \arg\max_{(h,r,t) \in \mathcal{G}} \cos((\mathbf{e}_h, \mathbf{r}, \mathbf{e}_t), \mathbf{q_i})) \\
\textbf{Community Filtering} &: \quad \arg\max_{\mathcal{C}_m \in \mathcal{K}} \cos(\mathbf{e}_{\mathcal{C}_m}, \mathbf{q_i}) \\
\textbf{DFS Path Traversal} &: \quad \mathcal{P}(q_i) = e_0 \xrightarrow{r_1} e_1 \xrightarrow{r_2} \cdots \xrightarrow{r_n} e_n \quad \text{s.t.} \quad \forall r_i \in \mathcal{R}, n \leq d
\end{aligned}
\tag{10}
$$

In general, the four retrieval paths exhibit distinct specialization patterns: ($i$) Entity Matching optimally handles single-hop simple queries requiring precise node identification, e.g., atomic fact check problem; ($ii$) Triple Matching dominates few-hop reasoning tasks by modeling $(h, r, t)$ compositional semantics, particularly effective for relationship inference; ($iii$) Community Filtering aims to address global queries, e.g., summarization and cross-domain problems through top-down filtering

in the cluster; (4) DFS Path Traversal scales to complex multi-constraint problems, we define the maximum depth $d = 5$. This specialization aligns with the cognitive spectrum from atomic facts to complex reasoning scenarios.

## A.4 EXPERIMENTAL ANALYSIS

### A.4.1 DATASETS

We firstly evaluate `Youtu-GraphRAG` in dual-mode on three widely used multi-hop QA datasets: HotpotQA (Yang et al. (2018)), MuSiQue (Trivedi et al. (2022)) and 2WikiMultiHopQA (abbreviated as 2Wiki Ho et al. (2020)), following the setting in (Jimenez Gutierrez et al. (2024); Gutiérrez et al. (2025)) for fair comparison.

To evaluate the framework's performance across diverse domains, we also employ GraphRAG-Bench(Xiao et al. (2025)), shorted as G-Bench, a benchmark dataset constructed from textbook corpora. Additionally, to prevent knowledge leaking, we propose two novel bilingual anonymous datasets, i.e., **AnonyRAG-CHS** and **AnonyRAG-ENG** and propose a challenging 'Anonymous Reversion' task.

We anonymize specific entity types (e.g., people, locations) in the dataset to break the model's memory shortcuts and prevent it from relying on pretrained knowledge rather than retrieved evidence. Moreover, we preserve semantic coherence through entity linking, enabling LLMs to maintain discourse comprehension despite anonymized mentions. The construction details of the dataset are documented in subsection A.7.

### A.4.2 BASELINES

We include three pipelines of research as baselines. ($i$) Naive RAG, as the standard RAG approach that retrieves top-$k$ document chunks using vector similarity search without any explicit knowledge structuring; ($ii$) Pure GraphRAG, which builds flat knowledge graphs for retrieval but lacks hierarchical organization, focusing primarily on relational reasoning through graph traversal algorithms, including GraphRAG (Edge et al. (2024)), LightRAG (Guo et al. (2024)), G-Retriever (He et al. (2024)) and HippoRAG 1&2 (Jimenez Gutierrez et al. (2024); Gutiérrez et al. (2025)); ($iii$) Tree-based GraphRAG, represents hierarchical methods that employ recursive clustering and summarization to construct multi-level knowledge trees including RAPTOR (Sarthi et al. (2024)) and $E^2$GraphRAG (Zhao et al. (2025)).

To ensure a fair performance comparison, we reproduce all the baselines and `Youtu-GraphRAG` with the same setting and evaluate with consistent metrics. In terms of base models, we maintain DeepSeek-V3-0324 and Qwen3-32B as the base LLMs and a lightweight embedding model all-MiniLM-L6-v2.

## A.5 RELATED WORK

While large language models (LLMs) demonstrate remarkable capabilities in language understanding and reasoning, they are known to be prone to hallucinations—generating confident yet factually incorrect outputs—especially when reasoning over complex or multi-hop queries (Dong et al., 2024b; Qin et al., 2024b; Kuang et al., 2025; Dong et al., 2024a; Qin et al., 2024a). Integrating LLMs with graph-structured knowledge, therefore, combines the generative flexibility of LLMs with the factual rigor of structured data, enabling more accurate and trustworthy reasoning over complex domains (Luo et al., 2023; Dong et al., 2023; Bei et al., 2025; Yasunaga et al., 2021; Luo et al., 2024). The approach of using LLM agents for graph-based reasoning was initiated by ToG (Sun et al., 2023), which explores graphs by sequentially expanding reasoning paths. Building upon this, ToG 2.0 (Ma et al., 2024) significantly refines the retrieval mechanism. It enables interactive access to both knowledge graphs and textual documents, fostering a context-sensitive reasoning process through the integration of multi-source information.

Evolving development of GraphRAG has progressed along two complementary research trajectories since the seminal work of (Edge et al., 2024). The first following approaches have evolved from LightRAG's (Guo et al., 2024) vector sparsification techniques to more sophisticated graph-aware methods. Subsequent innovations include GNN-RAG and GFM-RAG (Mavromatis & Karypis,

2024; Luo et al., 2025), which employ graph neural networks for enhanced node matching, and HippoRAG 1&2 (Jimenez Gutierrez et al., 2024; Gutiérrez et al., 2025) that introduced memory mechanisms and personalized PageRank algorithms for context-aware retrieval. While another group of methods have focused on improving the quality of knowledge organization, hierarchical approaches like RAPTOR (Sarthi et al., 2024) and $E^2$GraphRAG (Zhao et al., 2025) employ tree-like clustering and recursive summarization to enhance semantic organization. However, current research remain constrained by their specialized optimizations, either focusing on retrieval or construction in isolation, and lack a unified design. This fragmentation limits their performance on complex reasoning tasks requiring tight integration of knowledge organization and retrieval capabilities, which makes it even harder to adjust the entire framework for generalizability especially when domain shifts occur. Our work bridges this gap by developing a holistic framework that jointly optimizes both aspects while maintaining graph foundation model properties.

## A.6 PROMPT TEMPLATES IN LLMS GENERATION

We present the prompt templates in A.6.1 and A.6.2, which designed to evaluate whether permitting LLMs to utilize its parametric knowledge within the RAG system affects performance. To minimize confounding factors, we employed minimalistic prompts that solely differentiate between the two modes.

### A.6.1 REJECT MODE

> Given the question and the extracted knowledge from different retrieval paths, please answer the question below. If the extracted knowledge is not enough to answer, please reject to answer.
>
> Question: {query}
>
> Extracted Knowledge: {context}
>
> Answer:

### A.6.2 OPEN MODE

> Given the question and the extracted knowledge from different retrieval paths, please answer the question below. If the extracted knowledge is not enough to answer, please answer it based on your own knowledge.
>
> Question: {query}
>
> Extracted Knowledge: {context}
>
> Answer:

## A.7 DATA COLLECTION AND PROCESSING

All raw data in this study are sourced from the original texts of four classic novels: Water Margin, Dream of the Red Chamber, Moby-Dick, and Middlemarch. The copyrights of all these works have entered the public domain, thus presenting no copyright issues. In selecting data sources, we pursued two key objectives: (1) Ensuring comprehensive multilingual evaluation coverage, while (2) Maintaining sufficient complexity in entity representations (e.g., persons, locations) to rigorously assess model capabilities. The basic statistical information of the dataset is in Table 4.

In our data processing methodology, we employed DeepSeek for entity extraction from the corpus, then the data chunks are anonymized with the extracted entities. Query-answer pairs were constructed by DeepSeek using queries from 2Wiki and MuSiQue as seed templates. Upon acquiring the question-answer pairs, we performed entity anonymization using the same anonymization dic-

tionary as applied to the corpus. This procedure ensures that LLMs cannot effectively leverage parametric memorized patterns from questions. A representative example of anonymized question-answer pairs is presented in Table 5. As clearly demonstrated, while LLMs could handle questions according to common sense knowledge, their performance significantly degrades when confronted with anonymized versions of these questions. This phenomenon forces LLMs to rely on retrieved contextual information rather than depending solely on their parametric knowledge.

To avoid the variance in evaluating subjective questions, we finally converted the questions into two formats:

**Anonymity Reversion**. We provide LLMs with anonymized question-answer pairs as context, requiring to infer and reconstruct the original entities that were anonymized. This task specifically assesses the model's ability to leverage contextual clues for entity recovery.

**Multiple Choice**. To diversify question types and ensure objective evaluation, a subset of questions was converted into multiple-choice format.

We then performed zero-shot filtering to verify model performance on these transformed questions. This design preserves the original assessment objectives of testing the LLM's contextual reasoning capabilities while guaranteeing answer objectivity and uniqueness. Crucially, it mitigates potential unreliability introduced by LLM-as-judge evaluation paradigms. Table 6 presents representative cases of these two question formats.

Table 4: Question Type and Difficulty Distribution Statistics

| Question Type | Difficulty Level | Chinese Literature | | English Literature | | Total |
|---|---|---|---|---|---|---|
| | | *Water Margin* | *Dream of Red Chamber* | *Moby-Dick* | *Middlemarch* | |
| **Objective** | Simple (≤4 hops) | 29 | 26 | 54 | 54 | 163 |
| | Complex (>4 hops) | 24 | 34 | 51 | 22 | 131 |
| **Subjective** | Simple (≤4 hops) | 65 | 50 | 51 | 116 | 282 |
| **(Anonymous Reversion)** | Complex (>4 hops) | 146 | 314 | 142 | 219 | 821 |
| **Total** | | **264** | **424** | **298** | **411** | **1397** |

Table 5: Comparisons of anonymized and Non-anonymized questions

| Lang | Type | Question | Zero-shot LLM Response | Val |
|---|---|---|---|---|
| CHS | Raw Query | 从史进拜师到最终离开少华山，期间共经历了哪些身份转变? | 从富家子弟→王进徒弟→庄主→强盗同伙→落草为寇，共经历四次身份转变。 | Correct |
| | Annoy Query | 从[PERSON#277]拜师到最终离开[LOCATION#759]，期间共经历了哪些身份转变? | 从拜师到离开[LOCATION#759]，[PERSON#277]关键身份转变可梳理如下：1. 入门弟子阶段，初始以普通学徒身份拜入师门，主要承担基础杂役工作… | Wrong |
| ENG | Raw Query | What does the narrator's reaction to Queequeg's smoking in bed reveal about their relationship? | The narrator's acceptance of Queequeg's smoking reveals their deepening friendship and shared comfort. | Correct |
| | Annoy Query | What does the [PERSON#1999]'s reaction to [PERSON#200]'s smoking in bed reveal about their relationship? | To analyze what [PERSON#1999]'s reaction to [PERSON#200]'s smoking in bed reveals about their relationship, we would need more context about the specific reaction and… | Wrong |

A.8 PROMPT TEMPLATES DURING EXTRACTION

Table 6: Final Question-Answer Formats

| Lang | Question | Ground Truth |
|------|----------|--------------|
| **Anonymity Reversion** | | |
| CHS | 请根据上下文对下面这段问答
```
Q: 在[PERSON#532]离开[LOCATION#526]后，他在哪个村庄的酒店中与[PERSON#277]重逢？这个村庄附近的山上盘踞着哪两位头领？
A: [PERSON#532]在[LOCATION#110]附近的酒店与[PERSON#277]重逢，该村庄附近的[LOCATION#535]上盘踞[PERSON#503]和[PERSON#4]两位头领。
```
中已经被匿名化处理的所有人名和地名等进行推理，判断出被匿名的原本内容是哪些。 | PERSON#532——鲁智深
PERSON#277——史进
PERSON #4——周通
PERSON#503——李忠
LOCATION#526——五台山
LOCATION#110——桃花村
LOCATION#535——桃花山 |
| ENG | Please read the following QA pairs
```
Q: What does [PERSON#200]'s story about the wedding feast reveal about cultural misunderstandings?
A: The story reveals how cultural misunderstandings, such as [PERSON#588] mistaking the punchbowl for a finger-glass, can arise from ignorance of local customs.
```
then for all anonymized Persons and Locations, perform inference to determine the original content that was anonymized. | PERSON#200——Queequeg
PERSON#588——captai |
| **Multiple Choice** | | |
| CHS | 海棠诗社成立时，[PERSON#315]给自己取的别号是什么？这个别号与她居住的哪个场所相关？

A. [LOCATION#340]；[LOCATION#625]老农
B. [LOCATION#340]；[LOCATION#340]隐士
C. [LOCATION#625]老农；[LOCATION#340]
D. [LOCATION#340]居士；[LOCATION#625]老农 | C. (李纨，稻香老农，稻香村) |
| ENG | Which two physical traits do [PERSON#1035] and her daughter [PERSON#445] share in common?

A. Straight hair and round faces
B. Curly hair and square faces
C. Wavy hair and oval faces
D. Short hair and triangular faces | B. (Mrs. Garth, daughter Mary) |

You are an expert information extractor and structured data organizer. Your task is to analyze the provided text and extract as many valuable entities, their attributes, and relations as possible in a structured JSON format.
Guidelines:
1. Prioritize the following predefined schema for extraction;

Schema: {schema}
2. Flexibility: If the context doesn't fit the predefined schema, extract the valuable knowledge as needed;
3. Conciseness: The Attributes and Triples you extract should be complementary and no semantic redundancy.

4. Do NOT miss any useful information in the context; 5. Output Format: Return only JSON.
Example Output:
- Attributes: Map each entity to its descriptive features.
- Triples: List relations between entities in (entity_mention1, relation, entity_mention2) format.
- Entity_types: Map each entity to its schema type based on the provided schema.
Schema Evolution: If you find new and important entity types, relation types, or attribute types that are valuable for knowledge extraction, include them in a "new_schema_types" field. Notably, the strict threshold of adding new schema considering both importance and similarity to the pattern in the existing schema is 0.9.

Chunk: {chunk}

Example Output:
{
"attributes":
{ "Stephen King": ["profession: author"]
},

"triples": [ ["Shawshank Redemption", "based on", "Rita Hayworth and Shawshank Redemption"], ["Shawshank Redemption", "directed by", "Frank Darabont"]
],

"entity_types":
{ "Stephen King": "person", "Shawshank Redemption": "creative_work", "Rita Hayworth and Shawshank Redemption": "creative_work", "Frank Darabont": "person"
},

"new_schema_types":
{ "nodes": ["Movie"], "relations": ["starring"], "attributes": ["genre"]
}
}

A.8.1 SHCEMA-GUIDED QUERY DECOMPOSITION

You are a professional question decomposition expert specializing in multi-hop reasoning. Given the following ontology and the question, decompose the complex question into 2-3 focused sub-questions and identify involved schema types.
CRITICAL REQUIREMENTS:
1. Each sub-question must be:
• Specific and focused on a single fact or relationship by identifying all entities, relationships, and reasoning steps needed
• Answerable independently with the given ontology
• Explicitly reference entities and relations from the original question
• Designed to retrieve relevant knowledge for the final answer
2. For simple questions (1-2 hop), return the original question as a single sub-question 3. Analyze the question and identify all schema types that might be involved 4. Only return a concise JSON object with sub_questions array and involved_types object.

Ontology:
{ontology}

Question:
{question}

Example for complex question:
Original: "Which film has the director died earlier, Ethnic Notions or Gordon Of Ghost City?"
Output:
{
"sub_questions": [
"sub-question": "Who is the director of Ethnic Notions?",
"sub-question": "Who is the director of Gordon Of Ghost City?",
"sub-question": "When did the director of Ethnic Notions die?",
"sub-question": "When did the director of Gordon Of Ghost City die?"
],
"involved_types": {
"nodes": ["creative_work", "person"],
"relations": ["directed_by"],
"attributes": ["name", "date"]
}
}

Example for simple question:
Original: "What is the capital of France?"
Output:
{
"sub_questions": [
"sub-question": "What is the capital of France?"
],
"involved_types": {
"nodes": ["location"],
"relations": ["located_in"],
"attributes": ["name"]
}
}

### A.8.2 ITERATIVE REFLECTION AND REASONING

> You are an expert knowledge assistant using iterative retrieval with chain-of-thought reasoning.
>
> Current Question:
> {current_query}
>
> Available Knowledge Context:
> {context}
>
> Previous Thoughts:
> {previous_thoughts}
>
> Step step: Please think step by step about what additional information you need to answer the question completely and accurately.
>
> Instructions:
> 1. Analyze the current knowledge context and the question
> 2. Think about what information might be missing or unclear
> 3. If you have enough information to answer, in the end of your response, write "So the answer is:" followed by your final answer
> 4. If you need more information, in the end of your response, write a specific query begin with "The new query is:" to retrieve additional relevant information
> 5. Be specific and focused in your reasoning
>
> Your reasoning:

### A.9 PROMPT TEMPLATES DURING GENERATION

> Given the question and the extracted knowledge from different retrieval paths, please answer the question below. If the extracted knowledge is not enough to answer, please answer it based on your own knowledge.
>
> Question: {query}
>
> Extracted Knowledge: {context}
>
> Answer:

