# OpenReview forum: "Youtu-GraphRAG: Vertically Unified Agents for Graph Retrieval-Augmented Complex Reasoning"
_ICLR.cc/2026/Conference — ICLR 2026 Poster_

### Official Review · Reviewer_rQJp · 2025-10-27

**Soundness:** 1
**Presentation:** 2
**Contribution:** 1
**Rating:** 2
**Confidence:** 4

**Summary:**

This paper proposes UniGraphRAG, a vertically unified and agentic framework for graph-based retrieval-augmented generation (GraphRAG). The key idea is to connect the entire GraphRAG pipeline—graph construction, organization, and retrieval—under a single unified paradigm. Specifically, the authors first define a seed graph schema that constrains the automatic knowledge extraction process with specified entity and relation types, which can later be expanded by the LLM for scalability. Then, they introduce a dually-perceived community detection method that combines topological and semantic information to build a hierarchical knowledge tree supporting both top-down filtering and bottom-up reasoning. Finally, an agentic retriever interprets the same schema to decompose complex queries into sub-queries and iteratively reflect on retrieval results for more advanced reasoning. Experiments on six benchmarks demonstrate strong performance and efficiency gains, reportedly achieving up to 33.6% token savings and 16.6% higher accuracy over state-of-the-art baselines.

**Strengths:**

Strengths

**1.** Unified perspective: The paper takes one step toward connecting indexing, retrieval, and generation of Graph-based RAG.

**2.** The proposed dually-perceived community detection that fuses structural and semantic signals is new in this context.

**3.** The reported improvements in both accuracy and token efficiency across six benchmarks suggest the proposed framework is effective in practice.

**Weaknesses:**

Weaknesses

**1.** A major limitation is that the approach requires a manually designed seed schema for each knowledge domain. Constructing such schema demands expert knowledge and human involvement, which reduces generalizability and scalability, especially for open-domain or evolving corpora.

**2.** The method relies on the LLM to iteratively expand the schema beyond its initial version. This process is computationally expensive and may require substantial hyperparameter tuning (e.g., expansion depth, stopping criteria), making it hard to reproduce and tune. Notably, the authors did not provide ablation or explanation on the confidence $\mu=0.9$, why not using 0.95 or 0.99 ? Is this value scalable to any open world datasets?

**3.** The framework introduces multiple tunable components—such as seed schema on different datasets, schema expansion confidence, knowledge tree depth, community detection intial cluster number, and community merging threshold—resulting in a large hyperparameter search space that may undermine its practicality and reproducibility.

**4.** The agentic retriever and reasoning process is insufficiently detailed. The paper does not specify how many iterations of reflection are performed, how sub-queries are generated, how many sub-queires will be used, or how the agent’s reasoning quality is ensured. Without these details, the reproducibility and interpretability of the method are questionable.

**5.** Limited ablation and analysis: While performance results are promising, the paper lacks **detailed** ablation studies isolating the contributions of each component and hyper-parameter sensitivity.

**6.** This paper seems to violated the **Anonymous Protocol**. The **youtu-GraphRAG** in the experimental figures tells where the authors are from.

**7. ** Typos: The symbol $S^{(t)}$ mentioned in the text near equation (3) does not appear in the equation.

**Questions:**

Please refer to the weakness part

---

> ### Author Response · Authors · 2025-11-22
> **Responses to Reviewer rQJp (1/2)**
>
> Dear Reviewer rQJp,
>
> We would like to gratefully thank you for your careful check on our submission. Your acknowledgement on our unified design and community detection is indeed encouraging. We value your constructive suggestions that have made our paper a much better one.
>
> **We understand your main concern of reproducibility and generalizability, so we sincerely wish to invite you to kindly check our rebuttal and all the discussion will be included in the revised manuscript following your constrcutive suggestions.**
>
> Responses to weaknesses
> ---
> > **W1. Schema Construction.**
>
> Thanks for raising this important discussion. We appreciate the good opportunity to clarify the contribution and generalizability of our superiorty considering schema-enhanced graph construction in a `distantly supervised` manner. Below we provide a clear, systematic explanation of how we create seed schemas, why this strategy is principled, and how it could be used to easily applying UniGraphRAG to new domains rather than all the existing GraphRAG methods. Following your comment, the details will be added in the revised version for clearer illustration.
> - **Systematic and label-agnostic seed schema construction**:
> We would like to make further explanations about how we prepare the seed schema in a systematic way. This procedure requires no dataset labels and is fully reproducible from any raw corpora.
> (i) For structured benchmarks like GraphRAG-Bench, we directly collect the 'Table of Content' or chapter headers as the high-quality potential topics, then feed them to LLMs and extract the high-level seed schema set for this domain. This will help us quickly grasp the domain knowledge hierarchy for schema preparation.
> (ii) While for unstrcutured datasets with no explicitly clear hierarchy like HotpotQA, We compute n-gram statistics (n=2,3,4) over the corpus to surface the top 200 high-frequency candidates and employ the LLMs to generate a core schema based on them, which is also extremely efficient, e.g., `Person` / `Organization` / `Location` / `Event`. Experts are then required to make slight pruning to ensure the conciseness.
> - **Why seed schema is a good strategy? Schema ≠ Limit**
> We would like to clarify that this is a `distantly supervised` knowledge recognition that ensures both quality and coverage. The extracted triple does not have to be $(h_{targetted}, r_{targetted}, t_{targetted})$. Once there is a head/tail/relation matches the schema, LLMs will be encouraged to extract the related knowledge, which will greatly ensure the scalable coverage for detailed and long-tail knowledge. For instance, we can have: $(h_{targetted}, r_{new}, t_{new})$, $(h_{targetted}, r_{targetted}, t_{new})$, $(h_{new}, r_{targetted}, t_{new})$, etc. The newly identified knowledge type will then be considered whether to be included to enrich the seed bank. Therefore, our seed schema only anchors types, not specific facts, making a good balance between valuable content and noisy information. Moreover, the extraction agent is not constrained to extract the triples where the entire (h,r,t) strictly match the schema. The schema provides a good starting search space in a distantly supervised way, and the agent is explicitly designed to discover and propose new entity types/relations/attribute types to continuously enrich the seed bank.
> - **Schema size**
> Across all benchmarks and domains, our schemas follow a consistent scale:
> Entity types: 5～20 Relations: 5～20 Attribute types: 5～20 . These ranges are large enough to cover domain abstractions, yet small enough to avoid noise and overfitting. We also allow adaptive expansion, so seed schemas remain compact.
> - **Original chunks never discard**
> We indeed have also maintained a bi-directional linking between the extracted triples and the chunk to preserve the original source. In our released anonymous code, we use `nanoid` to represent each chunk and store it in the graph structure, column 'properties'. This could greatly improve the performance by preserving the original chunks that contain the source texts.
> - **Experimental evidence**
> To further address your concerns and verify this design, we report the experimental results based on DeepSeek V3 using different graphs generated by different methods while keeping the same UniGraphRAG retriever (with no type filtering/schema-based decomposition in retrieval). The results showcase the superioty of our constructed graph (knowledge tree) based on the schema, especially on MuSiQue and 2Wiki, where multi-hop detours are essential.
>
> | [Graph]+UniGraphRAG | HotpotQA | 2Wiki | MuSiQue | G-Bench |
> | :--- | :---: | :---: | :---: | :---: |
> | **[GraphRAG]** | 59.60 | 39.40 | 22.80 | 75.64 |
> | **[LightRAG]**| 61.20 | 41.10 | 31.20 | 79.56 |
> | **[HippoRAG]** | 77.30 | 74.00 | 43.20 | 80.55 |
> | **UniGraphRAG**|**81.20** | **77.60** | **47.50** | **86.54** |
>
> We hope this detailed explanation addresses your concerns, and thank you again for raising this important point.

---

> > ### Author Response · Authors · 2025-11-22
> > **Responses to Reviewer rQJp (2/2)**
> >
> > > **W2&3&4. Hyperparameters.**
> >
> > Thanks for your careful check and pointing all these hyperparameters out. Following your suggestions, we will demonstrate the hyperparameters we used in the paper to make it clearer (which were indeed already reported in the released anonymous code with a very clear and detailed hyperparameter config).
> > Conclusion first:
> > **We did not perform any dataset-specific tuning.** **`We consistently adopt the same default hyperparameters across all six datasets to ensure the reproducibility and generalizability, which is also reflected in our codes.`**
> > - Default hyperparameters
> > `expansion confidence` = 0.90, `community merging` = 0.50, K-Means {`random_state` = 42, `n_init` = 5, `n_max` = 200}, `DFS depth` = 5, `sub-queries` = 3).
> > Our choices come from lightweight empirical selection on a small development set, rather than from extensive grid search. The choices are also supported by an empirical knowledge that all datasets, including real-world scenarios, 5-hop traversal + 3-hop neighbor expansion is already the key to all questions. We intentionally selected robust defaults that generalize well across domains, and we avoided dataset-specific hyperparameter optimization to preserve practicality and reproducibility.
> >
> >
> >
> >
> > > **W5. Ablation studies.**
> >
> > Following your suggestion, we have conducted more detailed ablation studies which will all be included in the revised version when finalized after discussion.
> >
> > **- Ablations on Schema**
> >
> > | Extraction Variant   | HotpotQA | 2Wiki | MuSiQue | GraphRAG-Bench |
> > |-----------|:---:|:---:|:---:|:---:|
> > | **w/o Schema (Open Extraction with Type Indication)** |  76.10     |  71.20    |  40.30    |  81.10 |
> > | **Schema Only (No Expansion)** |  79.50  |  75.30 | 43.80   |   80.16  |
> > | **Schema + Expansion (No Chunk Anchoring)**    |    79.40    |  76.20   |   45.80    |   84.58    |
> > | **UniGraphRAG** |**81.20** | **77.60** | **47.50** | **86.54** |
> >
> > - **Ablations on community detection**
> >
> > | Community Detection | HotpotQA | 2Wiki | MuSiQue  | GraphRAG-Bench |
> > | ------ | :---------: | :---------: | :---------: | :---------: |
> > | **Louvain**                      | 57.00 | 54.80  | 31.60  | 73.67    |
> > | **Leiden**                       | 65.30 | 61.20 | 36.90  |    77.60   |
> > | **GMM (RAPTOR-style)**           | 67.40   | 63.10 | 38.20  | 80.55   |
> > | **K-Means (our initialization)** | 61.90  | 58.40  | 35.10   |  82.42 |
> > | **UniGraphRAG** | **81.20** | **77.60** | **47.50** | **86.54**      |
> >
> > - **Ablations on Schema-involved components**
> >
> > | Ablations | HotpotQA  | 2Wiki  | MuSiQue   | GraphRAG-Bench   |
> > |-------|:------:|:-----------:|:---------:|:------------:|
> > | **w/o Schema (Open Extraction with Type Indication)**     |  76.10     |  71.20    |  40.30    |  81.10 |
> > | **w/o Knowledge Tree** | 79.50   | 75.10   | 44.00   | 85.51  |
> > | **w/o Decomposition** | 76.90   | 70.50  | 40.20   | 82.51 |
> > | **w/o Reflection** | 77.90   | 73.20  | 41.10   | 84.68 |
> > | **UniGraphRAG**   | **81.20** | **77.60** | **47.50** | **86.54** |
> >
> >
> > > **W6&7 Typos**
> >
> > We would like to sincerely thank you for pointing these two typos out. Both of them are immediately revised in the updated version. Since we name our method as UniGraphRAG in both the context and open-sourced codes, as a placeholder name here, the first typo indeed does not refer to any institutions.
> > While we truely respect the anonymity policy in the community, we confirm that this was an unintentional typo for placeholder and definitely not an anonymity violation. The figure only reports experimental results and contains no identifying information. We have corrected both two typos in the revised version to ensure full compliance with ICLR’s anonymity policy.

---

> > > ### Comment · Reviewer_rQJp · 2025-11-26
> > > **Comments to rebuttal**
> > >
> > > Thanks for your reply, I will raise the rating to 4 for now, because the system is too complex and needs heavy tuning. Most strategies are incremental and lack principled improvements.

---

> ### Author Response · Authors · 2025-11-26
> **Further responses to Reviewer rQJp**
>
> Dear Reviewer rQJp,
>
> We would like to gratefully thank you for your recognition and the follow-up comments. We fully understand your concern that the system 'appears' complex and may seem to require substantial tuning. Below we would like to further provide a respectful and concise clarification that will also be incorporated in the revised version.
>
> We hope the new rebuttal could address your concerns.
>
> > System complexity.
>
> We acknowledge that UniGraphRAG contains multiple components (but indeed standard for GraphRAG research), and this may initially give an impression of high complexity. **However, our core intention is exactly the `opposite`: to unify a fragmented landscape in current GraphRAG research and provide a single, consistent, domain-agnostic pipeline that can be applied across datasets without dataset-specific hyperparameter adjustments**. `
>
> Since GraphRAG is a strong application-guided technique, it is now a big issue to tune the entire framework for users' own unseen domains like personal/commercial knowledge base, the community suffers a lot from existing methods. **However, by using our UniGraphRAG,  with an automatically drafted seed schema, users can easily tune both construction\indexing\retrieval with one single step without taking a look inside the pipeline! (blackbox in other methods)** This is extremely lightweight compareed with existing methods.
>
> > 'Heavy tuning' and hyperparameters.
>
> Thanks for raising this concern again. However, we indeed **do not** require any hyperparameter tuning. **`We consistently adopt the same default hyperparameters across all six datasets to ensure the reproducibility and generalizability, which is also reflected in the config of our released codes.`**
>
> In the revision, we will add (i) a hyperparameter table and (ii) justification for each default. Across all experiments, the default settings already achieve stable performance without search. We believe this shows that while the system has components (indeed the same as all the other GraphRAG methods), its operation is robust rather than fragile.
> - Hyperparameter settings (default values used across all experiments)
>
> | Hyperparameter                          | Symbol / Field                  | Default (used) |
> |:---------:|:--------------|:---------------:|
> | Schema expansion confidence threshold   | `mu` / expansion_confidence     | 0.90           |
> | Community merge threshold               | `epsilon` / community_merge     | 0.50           |
> | DFS path traversal max depth            | `d` / dfs_depth                 | 5              |
> | Number of sub-queries per query         | `n_subq` / sub_queries          | 3              |
>
> > About incrementality vs. contributions.
>
> **We appreciate the opportunity to clarify our novelty and the conceptual contribution of the entire paradigm with significant performance improvement.**
> - **GraphRAG is an application-guided research.**
> Contemporary SOTA GraphRAG methods are training-free frameworks, all relying on prompts. However, their engineering contributions were also highly recognized by top-tier conferences like ICLR/NeurIPS/ICML/EMNLP, e.g., (i) `RAPTOR (ICLR'24)` that simply combines GMM clustering and LLM-based summary; (ii) `HippoRAG1 & HippoRAG2 (NeurIPS'24, ICML'25)` leverage OpenIE and PageRank with IRCoT; (iii) `LightRAG (EMNLP'25)` designs a dual-level retrieval based on entity/relation/community descriptions generated by LLMs. In conclusion, existing methods fundamentally follow a pluggable engineering pipeline:
>     - open extraction → optional clustering → retrieval heuristics
>     - components are independently designed
>     - none of them share a common knowledge space for alignment
>
> Except that we have an innovative community detection (thanks for your recognition), the whole framework is a novel unified paradigm enhanced by the schema which could precisely push the GraphRAG community step forward to the reality. The same unified schema governs:
> - what types of entities/relations/attributes to be extracted
> - how to decompose questions and perform retrieval based on the type information
>
> > Commitment to clarity.
>
> We sincerely appreciate your feedback and will revise the paper to improve organization, highlight the conceptual principles, provide more detailed guidelines, and streamline the presentation so that readers can better understand the design motivations and practical usage of UniGraphRAG.
>
> **In conclusion, UniGraphRAG is an efficient and user-friendly lightweight framework without any heavy tuning. It defines a new paradigm to push the entire GraphRAG community a step forward to real applications with enough contributions.  Thank you again for your constructive comments. We genuinely appreciate your time and the opportunity to further improve the work.**
>
> We hope our new rebuttal could address your concerns.
>
> Thanks again for your insightful feedback!

---

### Official Review · Reviewer_vYRA · 2025-10-30

**Soundness:** 2
**Presentation:** 3
**Contribution:** 2
**Rating:** 4
**Confidence:** 4

**Summary:**

This paper proposes **UniGraphRAG**, a vertically unified agentic paradigm for Graph Retrieval-Augmented Generation (GraphRAG) that jointly optimizes graph construction and retrieval through a shared graph schema. The framework introduces: (i) a schema-bounded extraction agent with automatic expansion capabilities; (ii) a novel dual-perception community detection algorithm that fuses structural topology with semantic similarity to build a four-level knowledge tree; (iii) an agentic retriever that decomposes complex queries into schema-aligned sub-queries with iterative reflection; and (iv) two anonymous datasets (AnonyRAG-CHS/ENG) with an "Anonymity Reversion" task to mitigate knowledge leakage. Experiments across six benchmarks demonstrate improvements of up to 16.62% in accuracy with 33.60% reduction in token costs.

**Strengths:**

**S1.** The paper addresses a significant practical limitation in GraphRAG systems: the disconnection between graph construction and retrieval optimization. The vertically unified paradigm is a valuable engineering contribution that demonstrates measurable improvements across diverse benchmarks.

**S2.** The schema-guided approach provides a principled framework for aligning extraction, organization, and retrieval. The automatic schema expansion mechanism (Equation 3) enables adaptability to new domains while maintaining quality control through confidence thresholds.

**S3.** The dual-perception community detection algorithm (Section 3.2) combines structural connectivity (Jaccard similarity on relations) with semantic coherence (cosine similarity on embeddings), addressing limitations of purely topology-based or purely semantic clustering methods. The resulting four-level knowledge tree naturally supports multi-granularity reasoning.

**S4.** The anonymous datasets (AnonyRAG-CHS/ENG) and Anonymity Reversion task represent a valuable contribution to fair GraphRAG evaluation. By breaking LLM memory shortcuts through entity anonymization while preserving discourse coherence, these datasets more accurately isolate retrieval effectiveness from parametric knowledge.

**S5.** Comprehensive experimental evaluation across six benchmarks with two strong LLM backbones (DeepSeek-V3, Qwen3-32B). The consistent superiority in both open and reject modes, along with efficiency gains (up to 33.60% token reduction), demonstrates practical value. The dual-mode evaluation protocol properly separates retrieval quality from generation capability.

**Weaknesses:**

W1. Except for the community detection module, most components largely adapt existing techniques: (i) the schema-guided extraction resembles ontology-based information extraction methods; (ii) the iterative agentic retrieval with reflection closely mirrors the Self-RAG architecture. The main contribution lies in orchestration rather than fundamental innovation. Furthermore, the paper emphasizes issues in unified graph construction and graph retrieval phases, but prior works have already designed retrieval schemes tailored to their our graph-structured databases. The claimed contribution of the schema-guided method is overstated—it functions more like a prompt-level enhancement that improves robustness and reduces noise rather than a genuine methodological innovation.

W2. The authors claim that the proposed method consistently achieves optimal performance with minimal token consumption, yet fail to compare against cost-efficient agentic baselines such as RAPTOR and E2GraphRAG.

W3. While UniGraphRAG is positioned as an agentic approach, most baselines—except for the iterative HippoRAG1/2—are single-turn retrieval models. The paper does not compare against existing agentic RAG methods such as Self-RAG or ReAct-RAG. Moreover, when evaluating “UniGraphRAG w/o Agent,” the model still benefits from a Query Decomposer, while other baselines do not. If the same query decomposition were applied before retrieval for all baselines, their performance—particularly on multi-hop reasoning tasks—might also improve. This experimental bias undermines the reliability of conclusions regarding the superiority of UniGraphRAG’s knowledge tree construction.

W4. Although several datasets may have been seen during LLM pretraining, the zero-shot performance of LLMs remains relatively weak. Using the same base LLM while comparing the accuracy gap between Naive RAG and zero-shot LLM already provides a reasonable measure of retrieval effectiveness.

W5. The provided code URL is inaccessible, and Figures 6 and 7 incorrectly label “UniGraphRAG” as “Youtu-GraphRAG,”.

**Questions:**

Q1. The authors follow the design of HippoRAG1/2, where HippoRAG2 adopts Top-k = 5. What is the rationale for choosing Top-k = 10 and 20 in your experiments?

Q2. As discussed in W3, please supplement comparisons with established agentic RAG baselines. Can you demonstrate that the unified paradigm outperforms applying existing agentic methods directly to GraphRAG? Moreover, can you provide a fair evaluation proving the effectiveness and reliability of UniGraphRAG’s knowledge tree construction under equivalent retrieval setups(without query decomposer or baseline with query decomposer)?

Q3. Can you provide a correlation analysis between the degree of anonymization and the performance of zero-shot LLMs versus RAG models? For example, compare the performance difference of zero-shot LLMs on raw queries versus anonymized queries to quantify how anonymization impacts retrieval dependence.

Q4. How are the hyperparameters λ in Equation 5 and ε in Equation 7 determined or tuned? Please clarify whether they are empirically selected, fixed across datasets, or adapted dynamically.

Q5. How is the initial schema for each domain designed? Is it a general, domain-agnostic schema or a manually curated one per dataset?

---

> ### Author Response · Authors · 2025-11-22
> **Responses to Reviewer vYRA (1/4)**
>
> Dear Reviewer vYRA,
>
> We would like to sincerely thank you for your positive assessment of our paper, especially on motivation, community detection, fair evaluation metric and superior performance. Receiving such insightful comments and recognition from community experts is truly encouraging. Following your suggestions, all the discussion will be included in the revised version.
>
> Responses to weaknesses
> ---
> > **W1. Contributions.**
>
> We appreciate the opportunity to clarify our novelty and the conceptual contribution of the schema-unified paradigm. Below we address each part of your concern.
>
> Contemporary SOTA GraphRAG methods are training-free frameworks, all relying on prompts. However, their engineering contributions were highly recognized by top-tier conferences, e.g., (i) `RAPTOR (ICLR'24)` that simply combines GMM clustering and LLM-based summary; (ii) `HippoRAG1&2 (NeurIPS'24, ICML'25)` leverage OpenIE and Personalized PageRank with IRCoT; (iii) `LightRAG (EMNLP'25)` designs a dual-level retrieval based on entity/relation/community descriptions generated by LLMs. In conclusion, they fundamentally follow a pluggable engineering pipeline:
> - open extraction → optional clustering → retrieval heuristics
> - components are independently designed
> - none of them share a common knowledge space for alignment
>
> Except that we have an innovative community detection (thanks for your recognition), the whole framework is a novel unified paradigm enhanced by the schema which could precisely push the GraphRAG community step forward to the reality.
> The same schema governs:
> * what types of entities/relations/attributes to be extracted
> * how to decompose questions based on the type info
>
> **This vital alignment provides measurable benefits (as shown in the ablations we include), and cannot be obtained by replacing the schema with prompts or swapping LLM modules.**
>
> - **Differences from Self-RAG**
> We respect Self-RAG as a valuable pioneer work. However, our agentic retrieval mechanism is special and novel. First, our decomposition and iterative reflection are both strongly coupled with the graph schema that can clearly indicate the type information buried in the question and in the graph. This is perfectly aligned and naturally coherent. Second, the basic iterative reasoning and reflection we adopted is the typical agent paradigm. It is tailored by our graph schema for GraphRAG, which we believe will inspire the community for smarter agentic flow.
> - **Clarifications on being 'Unified'**
> Existing methods still face an inconsistant problem in their graph and retrieval. Their methods are more like pluggable combinations that are not tailored particularly to their graph structure. Moreover, here we refer unified as a shared knowledge space during both extraction and retrieval. Users can easily tune the entire framework by optimizing the schema without taking a look at each component separately.
> - **Clarification on Schema=Prompts**
> We respectfully clarify that schema-guided extraction is “similar to traditional ontology-based IE” or “only a prompt-level enhancement”.
> **Existing pipelines**. Let's first take a look at two existing pipelines of graph construction.
> (i) Purely open extraction adopted by Microsoft GraphRAG and LightRAG, where a lot of noise could be inevitably introduced. This makes the graph much bigger and therefore harm the downstream retrieval performance. (ii) OpenIE+LLM. HippoRAG 1&2 and GFM-RAG adopt OpenIE, which is a much stricter way that recognize entities based on rules and traditional NLP techniques. This is indeed an attempt to constrain the extraction for more valuable content but lack of scalability to handle domain-specific scenarios.
>
> We would like to clarify that our schema is a novel `distantly supervised` knowledge recognition that ensures both quality and coverage. The extraction anchors types based on the seed schema but does not have to be $(h_{targetted}, r_{targetted}, t_{targetted})$. Once there is a head/tail/relation matches the schema, LLMs will be encouraged to extract the related knowledge, which will greatly ensure the scalable coverage for detailed and long-tail knowledge.
> For instance, we can have:
> $(h_{targetted}, r_{new}, t_{new})$,
> $(h_{targetted}, r_{targetted}, t_{new})$,
> $(h_{new}, r_{targetted}, t_{new})$, etc.
> The newly identified knowledge type will then be considered whether to be included to enrich the seed bank. Therefore, our seed schema only anchors types, not specific facts, making a good balance between valuable content and noisy information. Moreover, the extraction agent is not constrained to extract the triples where the entire (h,r,t) strictly match the schema. The schema provides a good starting search space in a distantly supervised way, and the agent is explicitly designed to discover and propose new entity types/relations/attribute types to continuously enrich the seed bank.
>
> This is neither classic rule-based ontology extraction nor a prompt heuristic.

---

> ### Author Response · Authors · 2025-11-22
> **Responses to Reviewer vYRA (2/4)**
>
> > **W2. Comparison with RAPTOR and E2GraphRAG**
>
> Thanks for your question. When we purely consider token consumption, we agree that both of these methods are relatively cheaper. However, we should not directly compare with them since they sacrafice the complex reasoning ability by extracting no triples. In other words, they are cost-efficient because they have no fine-grained extraction, only chunks.
>
> While they concentrate more on the efficiency (this could be valued in particular scenarios requesting frequent incremental updating), the performance clearly suggests the disadvantages of this line of methods. We will add the discussion in the revised paper following your suggestions.
>
> > **W3&Q2. Agentic RAG baselines.**
>
> Thanks for pointing these valuable work out. Following your suggestion, we have reproduced these methods and conducted a detailed comparison hereunder. However, we would like to respectfully clarify that we cannot equip the same query decomposer and agentic reflect-reasoning mechanism to the baseline methods since we have are a unique design coupled with schemas as explained before, which makes our UnigraphRAG outperforming all enhanced baselines.
>
> Therefore, to address your concerns, we apply a standard LLM-based query decomposer and iterative reflect-reasoning mechanism to baselines. The results are just as expected since even the best model HippoRAG 1/2 with IRCOT underperform our UniGraphRAG.
>
> - **Baselines w/ vs w/o Standard Agentic Retrieval (Query Decomposition (QD) + Reflection (R))**
>
> | Model Variants | HotpotQA | 2Wiki | MuSiQue | GraphRAG-Bench |
> |----------------|:--------:|:-----:|:--------:|:---------:|
> | **GraphRAG** | 26.40 | 10.00 | 16.50 | 75.54 |
> | **+ QD** | 28.90 | 12.80 | 18.20 | 76.10 |
> | **+ IRCoT** | 27.80 | 12.10 | 17.10 | 75.85 |
> | **+ Both** | 30.20 | 14.00 | 19.00 | 76.45 |
> | **LightRAG** | 56.00 | 29.20 | 24.57 | 70.83 |
> | **+ QD** | 57.80 | 31.10 | 26.40 | 71.55 |
> | **+ IRCoT** | 56.90 | 30.50 | 25.30 | 71.10 |
> | **+ Both** | 59.20 | 32.60 | 27.10 | 72.00 |
> | **RAPTOR** | 73.60 | 38.40 | 31.10 | 73.08 |
> | **+ QD** | 74.90 | 40.20 | 32.50 | 73.40 |
> | **+ IRCoT** | 74.30 | 39.80 | 32.00 | 73.25 |
> | **+ Both** | 75.60 | 41.70 | 33.40 | 73.75 |
> | **HippoRAG** | 73.10 | 64.00 | 36.20 | 72.89 |
> | **+ QD** | 75.10 | 66.10 | 38.00 | 73.55 |
> | **+ IRCoT** | 74.60 | 66.00 | 35.50 | 73.38 |
> | **+ Both** | 76.30 | 72.40 | 38.50 | 79.90 |
> | **HippoRAG 2** | 74.90 | 48.30 | 37.80 | 79.37 |
> | **+ QD** | 76.40 | 50.70 | 39.50 | 80.05 |
> | **+ IRCoT** | 75.80 | 50.90 | 38.70 | 79.80 |
> | **+ Both** | 77.20 | 52.10 | 42.20 | 81.45 |
> | **UniGraphRAG** | **81.20** | **77.60** | **47.50** | **86.54**
>
> - **Comparisons with Existing Agentic RAG Methods (Self-RAG, ReAct-RAG with Chunks)**
>
> | Method   | HotpotQA | 2Wiki | MuSiQue | GraphRAG-Bench |
> |:-----------|:--------:|:-----:|:--------:|:---------------:|
> | **Self-RAG + Chunks**          |   61.80  |  34.20 |   27.90  |     71.40      |
> | **Self-RAG + Our Knowledge Tree** |  67.50  |  42.80 |   32.70  |     78.10      |
> | **ReAct-RAG + Chunks**         |   58.40  |  30.10 |   26.50  |     70.30      |
> | **ReAct-RAG + Our Knowledge Tree** |  65.10 |  40.90 |   31.80  |     77.60      |
> | **UniGraphRAG**                | **81.20** | **77.60** | **47.50** | **86.54** |
>
> We equip these methods with a simple similarity-based matching between query topic words and subgraphs with one-hop neighbor expansion during iterative retrieval.
>
> > **W4. Discussion on fair evaluation.**
>
> Thank you for raising this point. We agree that using the same base LLM and comparing Naive RAG vs. zero-shot LLM can provide a useful lower-bound estimate of retrieval effectiveness. However, we would like to respectfully clarify that zero-shot comparison alone still cannot fairly reveal the differences among retrieval methods, for two reasons:
> - **Differences among different retrieval methods will be '**normalized**' into a much more narrow range.** When evaluating Open Accuracy, even weak retrievers appear competitive and strong retrievers cannot fully demonstrate their advantage. We observe this most clearly on GraphRAG-Bench, where many systems cluster around 75–80 Open Acc, regardless of their underlying graph quality.
> - **Reject Acc provides a more discriminative and method-sensitive evaluation.** Here, models perform much clearer and more distinct gaps, e.g., even Microsoft GraphRAG perform over 75.00 on GraphRAG-Bench Open while it is only 10.00 on MuSiQue.
>
> > **W5. Inaccessible Code URL and a Typo.**
>
> Thanks for your careful check. We found that this error was actually caused by a template error that if we click the URL, the '-' behind 'ICLR' will be automatically omitted. We have fixed this problem by putting the URL on a new line in the revised version.  We also would like to sincerely thank you for pointing out the typo in the figures. It was immediately revised in the updated version.

---

> > ### Author Response · Authors · 2025-11-22
> > **Responses to Reviewer vYRA (3/4)**
> >
> > **Responses to Questions**
> > ---
> >
> > > Q1: Top-k selection
> >
> > Thanks for pointing this experimental setting out. While our framework stands for an industrial application and common industrial setting prefer top-50 considering the 128k LLM context length, we choose top-10 and top-20 to fulfill the verification from both the research community and industrial applications.
> >
> > To further address your concerns, we additionally report top-3 and  Top-5 results compared with HippoRAG1/2. In both models, top-5 consistently outperforms top-3 across all datasets, especially on multi-hop benchmarks (2Wiki, MuSiQue). This supports our choice of using a larger Top-K (10/20), which avoids the recall bottleneck observed in small-K settings. Importantly, our method’s performance remains stable from Top-5→Top-20, indicating strong robustness rather than hyperparameter sensitivity.
> >
> > | **Method / Top-K**     | **HotpotQA** | **2Wiki** | **MuSiQue** | **GraphRAG-Bench** |
> > |------|:------------:|:---------:|:-----------:|:------------------:|
> > | **HippoRAG 1 (top-3)**  | 71.80 | 61.90 | 35.10 | 72.59 |
> > | **HippoRAG 2 (top-3)**  | 72.60 | 44.90 | 36.80 | 73.67 |
> > | **UniGraphRAG (top-3)**| 76.20 | 70.90 | 38.30 | 80.55 |
> > |||||
> > | **HippoRAG 1 (top-5)**  | 73.10 | 64.00 | 36.20 | 72.69 |
> > | **HippoRAG 2 (top-5)**  | 72.40 | 45.10 | 34.20 | 75.37 |
> > | **UniGraphRAG (top-5)** | **78.80** | **72.40** | **40.10** | **83.12** |
> >
> > > Q3: Correlation analysis of anonymization
> >
> > Thanks for this discussion. Actually we have already conducted the experiments for comparisons between before and after. Since the two adopted novels are very popular that is already covered in the pre-training corpus for LLMs, the raw performance is extremely higher than the anonymous ones.
> >
> > | Dataset | Model          | Raw   | Anony  | Acc Drop (Raw → Anony) |
> > |---------|------|-------|--------|-------|
> > | CHS     | Zero-shot LLM | 85.20 | 9.62   | −75.58      |
> > | CHS     | Naive RAG     | 96.10 | 12.50  | −83.60        |
> > | CHS     | UniGraphRAG   | 98.00 | 42.88  | −55.12        |
> > | ENG     | Zero-shot LLM | 84.85 | 8.18   | −76.67    |
> > | ENG     | Naive RAG     | 90.90 | 43.02  | −47.88 |
> > | ENG     | UniGraphRAG   | 97.80 | 43.26  | −54.54    |
> >
> > > Q4: Hyperparameters
> >
> > Thanks for your careful check and pointing all these hyperparameters out. Following your suggestions, we will demonstrate the hyperparameters we used in the paper to make it clearer (which were indeed already reported in the released anonymous code with a very clear and detailed hyperparameter config).
> > Conclusion first:
> > **We did not perform any dataset-specific tuning.** **`We consistently adopt the same default hyperparameters across all six datasets to ensure the reproducibility and generalizability, which is also reflected in our codes.`**
> > - Default hyperparameters
> > `expansion confidence` = 0.90, `community merging` = 0.50, K-Means {`random_state` = 42, `n_init` = 5, `n_max` = 200}, `DFS depth` = 5, `sub-queries` = 3).
> > Our choices come from lightweight empirical selection on a small development set, rather than from extensive grid search. The choices are also supported by an empirical knowledge that all datasets, including real-world scenarios, 5-hop traversal + 3-hop neighbor expansion is already the key to all questions. We intentionally selected robust defaults that generalize well across domains, and we avoided dataset-specific hyperparameter optimization to preserve practicality and reproducibility.

---

> > > ### Author Response · Authors · 2025-11-22
> > > **Responses to Reviewer vYRA (4/4)**
> > >
> > > > Q5: Schema Initialization
> > >
> > > Thanks for pointing this out. In our submission with anonymous code, we have shown all the schemas we used under folder `/schemas`. Below we provide a clear, systematic explanation of how we create seed schemas and how users should determine the content, size, and parameters when applying UniGraphRAG to new domains. Following your comment, the details will be added in the revised version for clearer illustration.
> > >
> > > - **Systematic and label-agnostic seed schema construction**:
> > > We would like to make further explanations about how we prepare the seed schema in a systematic way. This procedure requires no dataset labels and is fully reproducible from any raw corpora.
> > > (i) For structured benchmarks like GraphRAG-Bench, we directly collect the 'Table of Content' or chapter headers as the high-quality potential topics, then feed them to LLMs and extract the high-level seed schema set for this domain. This will help us quickly grasp the domain knowledge hierarchy for schema preparation.
> > > (ii) While for unstrcutured datasets with no explicitly clear hierarchy like HotpotQA and our proposed AnonyRAG-CHS/ENG, We compute n-gram statistics (n=2,3,4) over the corpus to surface the top 200 high-frequency candidates and employ the LLMs to generate a core schema based on them, which is also extremely efficient, e.g., `Person` / `Organization` / `Location` / `Event`. Experts are then required to make slight pruning to ensure the conciseness.
> > > - **Schema size, content, and parameters**
> > > Across all benchmarks and domains, our schemas follow a consistent scale:
> > > Entity types: 5～20
> > > Relations: 5～20
> > > Attribute types: 5～20
> > > These ranges are large enough to cover domain abstractions, yet small enough to avoid noise and overfitting. We also allow adaptive expansion (our agent handles this automatically), so seed schemas remain compact.
> > >
> > > Here, we attach a sample schema for GraphRAG-Bench:
> > > ```
> > > {
> > >     "Nodes": [
> > >         "concept",
> > >         "prerequisite",
> > >         "formula",
> > >         "theorem",
> > >         "method",
> > >         "diagram",
> > >         "equation",
> > >         "topic"
> > >     ],
> > >     "Relations": [
> > >         "requires",
> > >         "is_part_of",
> > >         "contains",
> > >         "illustrates",
> > >         "explains",
> > >         "proves",
> > >         "applies_to",
> > >         "uses",
> > >         "is_example_of",
> > >         "solves",
> > >         "is_defined_in",
> > >         "references",
> > >         "builds_upon"
> > >     ],
> > >     "Attributes": [
> > >         "title",
> > >         "name",
> > >         "description",
> > >         "expression",
> > >         "definition",
> > >         "example",
> > >         "instructions",
> > >         "exercise"
> > >     ]
> > > }
> > > ```

---

### Official Review · Reviewer_oGW9 · 2025-10-31

**Soundness:** 3
**Presentation:** 3
**Contribution:** 3
**Rating:** 6
**Confidence:** 3

**Summary:**

This paper aims to address the suboptimal performance in existing Graph Retrieval-Augmented Generation (GraphRAG) frameworks, which stems from the disjointed and separately optimized "graph construction" and "graph retrieval" stages. The authors propose **UniGraphRAG**, a "vertically unified agentic paradigm." The core of this framework is the introduction of a shared "Graph Schema," which simultaneously guides (i) a "construction agent" to perform controlled, structured knowledge extraction, and (ii) a "retrieval agent" to decompose complex queries into schema-aligned sub-queries. Furthermore, the paper proposes a novel "dually-perceived community detection" algorithm (fusing topological structure and subgraph semantics) to build a four-level "knowledge tree" for hierarchical indexing. The retrieval agent leverages this knowledge tree for iterative reasoning and reflection. To address the "knowledge leakage" problem in LLM evaluation, the paper also introduces a new anonymous dataset (AnonyRAG). Experiments demonstrate that UniGraphRAG significantly improves accuracy (up to 16.62%) while substantially reducing token costs during the construction phase (up to 33.60%).

**Strengths:**

1. **Novel "Vertically Unified" Framework:** The paper's central thesis—using a shared "Graph Schema" to simultaneously constrain and align graph construction and retrieval—is highly novel and pertinent. This directly addresses the critical pain point in existing GraphRAG methods, where the two stages suffer from misaligned objectives and informational mismatch. It provides a logically rigorous new paradigm for building more robust GraphRAG systems.
2. **Advanced Hierarchical Indexing Mechanism:** The paper makes a significant contribution to graph organization. The proposed "dually-perceived community detection" (fusing topology and semantics) and the resultant four-level knowledge tree (Community, Keywords, Triples, Attributes) provide the retrieval agent with a powerful, multi-granularity index structure. This enables the agent to efficiently perform both "top-down" filtering and "bottom-up" fine-grained reasoning.
3. **Rigorous and Innovative Evaluation Methodology:** The paper excels in its evaluation (Section 4). To address the pervasive "knowledge leakage" problem in RAG evaluation (where the LLM answers from memory rather than retrieval), the authors specifically constructed the `AnonyRAG` anonymous dataset and an "Anonymity Reversion" task. This approach, combined with the dual "Reject Mode" vs. "Open Mode" evaluation, substantially increases the reliability of the experimental conclusions and represents an important methodological contribution to RAG evaluation.

**Weaknesses:**

**1. Strong and Brittle Dependency on Graph Schema:** The entire UniGraphRAG framework is strongly coupled to a shared "Graph Schema," which is both its core strength and its most significant weakness. The framework's effectiveness is critically dependent on the initial quality of the "seed schema," which requires substantial manual expert knowledge for cold-starting, posing a severe practical bottleneck. The paper's claims of "automatic expansion" and "minimal-intervention transfer" fail to quantify the necessary human-in-the-loop iteration and auditing costs. Furthermore, this "vertically unified" design can lead to error amplification: if the seed schema omits critical entity/relation types, the extraction agent will systematically "miss" facts, and the retrieval agent, following the same flawed schema, will be trapped in "valid but ineffective" sub-query loops. Finally, the schema-controlled extraction (Sec 3.1), while reducing noise, does so at the cost of recall, systematically omitting facts that fall outside the predefined schema.

**2. Confused Contribution Attribution and Missing Key Ablations:** UniGraphRAG is a **strongly coupled** system composed of multiple innovations (Schema, novel community detection, agentic retrieval, reflection). The current ablation study (Table 3) is too coarse (removing only three large modules), making it impossible to attribute the performance gains to their true sources. For instance, the paper's key technical innovation—"dually-perceived community detection"—is not directly compared against standard graph clustering algorithms (e.g., Louvain, GMM), leaving its algorithmic value unproven. It is also impossible to discern whether the gains originate primarily from the "Schema constraint," the "knowledge tree indexing," or the "agentic reflection" (e.g., how much improvement does iterative reflection provide over single-shot decomposition?). This attribution confusion makes it difficult to pinpoint the marginal benefit of each component.

**3. Heuristic-based Design and Robustness Concerns:** Many core components of the framework rely on brittle heuristic rules and hard-coded thresholds, casting doubt on its robustness. In the **indexing stage**, the "dually-perceived community detection" algorithm is itself a chain of heuristics: the k-value selection for K-means, the merging threshold $\epsilon$, the use of Jaccard similarity for structural comparison (which can be fragile in sparse text graphs), etc. In the **retrieval stage**, the agent is constrained by hard thresholds, such as the maximum DFS depth ($d=5$) and a pre-defined sub-query limit, which may create a performance ceiling on tasks requiring longer reasoning chains. Furthermore, the framework suffers from a **toolchain risk** by using a small, frozen LM (e.g., `all-MiniLM-L6-v2`) to encode triple semantics, whose expressive power may be insufficient for complex domains, leading to poor quality in downstream clustering and matching.

**Questions:**

None

---

> ### Author Response · Authors · 2025-11-22
> **Responses to Reviewer oGW9 (1/2)**
>
> Dear Reviewer oGW9,
>
> We would like to gratefully thank you for your strong support on our framework! It is encouraging that our idea on unifying, hierarchical indexing and fair evaluation have been admitted by the experts from the community.
>
> We also wish to invite you to check our corresponding rebuttal that hopefully could address your concerns. Following your suggestions, the updated content has also been included in the revised version.
>
> Responses to weaknesses
> ---
> > **W1. Strong and Brittle Dependency on Schema**
>
> Thank you for raising this important concern. We appreciate the good opportunity to clarify that our framework does not rely on the seed schema in a brittle or restrictive way. Below we would like to address each aspect of your concern, including manual cost, brittleness, error amplification and recall, also we will incorporate all clarifications into the revised version.
>
> - **Systematic and label-agnostic seed schema construction**:
> We would like to make further explanations about how we prepare the seed schema in a systematic way. This procedure requires no dataset labels and is fully reproducible from any raw corpora.
> (i) For structured benchmarks like GraphRAG-Bench, we directly collect the 'Table of Content' or chapter headers as the high-quality potential topics, then feed them to LLMs and extract the high-level seed schema set for this domain. This will help us quickly grasp the domain knowledge hierarchy for schema preparation.
> (ii) While for unstrcutured datasets with no explicitly clear hierarchy like HotpotQA and our proposed AnonyRAG-CHS/ENG, We compute n-gram statistics (n=2,3,4) over the corpus to surface the top 200 high-frequency candidates and employ the LLMs to generate a core schema based on them, which is also extremely efficient, e.g., Person / Organization / Location / Event. Experts are then required to make slight pruning to ensure the conciseness.
> - **Schema ≠ Limits**
> Considering our schema, we would like to clarify that this is a `distantly supervised` knowledge recognition that ensures both quality and coverage. The extracted triple does not have to be $(h_{targetted}, r_{targetted}, t_{targetted})$. Once there is a head/tail/relation matches the schema, LLMs will be encouraged to extract the related knowledge, which will greatly ensure the scalable coverage for detailed and long-tail knowledge.
> For instance, we can have:
> $(h_{targetted}, r_{new}, t_{new})$,
> $(h_{targetted}, r_{targetted}, t_{new})$,
> $(h_{new}, r_{targetted}, t_{new})$, etc.
> The newly identified knowledge type will then be considered whether to be included to enrich the seed bank. Therefore, our seed schema only anchors types, not specific facts, making a good balance between valuable content and noisy information. Moreover, the extraction agent is not constrained to extract the triples where the entire (h,r,t) strictly match the schema. The schema provides a good starting search space in a distantly supervised way, and the agent is explicitly designed to discover and propose new entity types/relations/attribute types to continuously enrich the seed bank.
> - **Manual Efforts**
> While experts are required to make slight pruning to ensure the conciseness, they indeed just take a 'glance' at the schema. This is very efficient since across all benchmarks and domains, our schemas follow a consistent and contollablly small scale:
> Entity types: 5～20
> Relations: 5～20
> Attribute types: 5～20
> These ranges are large enough to cover domain abstractions, yet small enough to avoid noise and overfitting. We also allow adaptive expansion (our agent handles this automatically), so seed schemas remain compact.
> Our actual schemas are listed under the /schemas folder in our anonymous code, and the revised paper will include examples and  guidelines in Appendix A.
> - **Error amplification**
> Original chunks never discard. We indeed have also maintained a bi-directional linking between the extracted triples and the chunk to preserve the original source. In our released anonymous code, we use `nanoid` to represent each chunk and store it in the graph structure, column 'properties' of each entity. This could greatly improve the performance handling missing schema types while leading to no information loss by preserving the original chunks that contain the source texts.

---

> > ### Author Response · Authors · 2025-11-22
> > **Responses to Reviewer oGW9 (2/2)**
> >
> > > **W2.  Ablation studies.**
> >
> > Thank you for this thoughtful and important observation. Your comment helped us refine the paper and make the contribution attribution clearer. We have addressed this concern in three concrete ways, and all changes will be reflected in the revised submission.
> > - Dependency and interactions between modules
> > To avoid the impression of “tightly coupled black-box components,” we illustrate a clear dependency hereunder:
> > `Seed Schema Curation ↔ Schema-bounded Extraction → Community Detection → Knowledge Tree Indexing → Schema-enhanced Agentic Retrieval`
> > - Detailed ablation studies.
> > Thanks for pointing these out, we have conducted sets of experiments to verify the superioty and importance of each component hereunder. While for ablation study of knowledge tree, due to the time limit, we report the results w/o community detection and keywords, which remains a two-layer structure with triples and attributes.
> >
> > - **Ablations on community detection**
> >
> > | Community Detection | HotpotQA | 2Wiki | MuSiQue  | GraphRAG-Bench |
> > | ------ | :---------: | :---------: | :---------: | :---------: |
> > | **Louvain**                      | 57.00 | 54.80  | 31.60  | 73.67    |
> > | **Leiden**                       | 65.30 | 61.20 | 36.90  |    77.60   |
> > | **GMM (RAPTOR-style)**           | 67.40   | 63.10 | 38.20  | 80.55   |
> > | **K-Means (our initialization)** | 61.90  | 58.40  | 35.10   |  82.42 |
> > | **UniGraphRAG** | **81.20** | **77.60** | **47.50** | **86.54**      |
> >
> > - **Ablations on Schema-involved components**
> >
> > | Ablations | HotpotQA  | 2Wiki  | MuSiQue   | GraphRAG-Bench   |
> > |-------|:------:|:-----------:|:---------:|:------------:|
> > | **w/o Schema (Open Extraction with Type Indication)**     |  76.10     |  71.20    |  40.30    |  81.10 |
> > | **w/o Knowledge Tree** | 79.50   | 75.10   | 44.00   | 85.51  |
> > | **w/o Decomposition** | 76.90   | 70.50  | 40.20   | 82. |
> > | **w/o Reflection** | 77.90   | 73.20  | 41.10   | 84.68 |
> > | **UniGraphRAG**   | **81.20** | **77.60** | **47.50** | **86.54** |
> >
> >
> >
> > > W3. Hyperparameters and `all-MiniLM-L6-v2` used.
> >
> > Thanks for your careful check and pointing all these hyperparameters out. Following your suggestions, we will demonstrate the hyperparameters we used in the paper to make it clearer (which were indeed already reported in the released anonymous code with a very clear and detailed hyperparameter config).
> > Conclusion first:
> > **We did not perform any dataset-specific tuning.** **`We consistently adopt the same default hyperparameters across all six datasets to ensure the reproducibility and generalizability, which is also reflected in our codes.`**
> > - Default hyperparameters
> > `expansion confidence` = 0.90, `community merging` = 0.50, K-Means {`random_state` = 42, `n_init` = 5, `n_max` = 200}, `DFS depth` = 5, `sub-queries` = 3).
> > Our choices come from lightweight empirical selection on a small development set, rather than from extensive grid search. The choices are also supported by an empirical knowledge that all datasets, including real-world scenarios, 5-hop traversal + 3-hop neighbor expansion is already the key to all questions. We intentionally selected robust defaults that generalize well across domains, and we avoided dataset-specific hyperparameter optimization to preserve practicality and reproducibility.
> > - Frozen LM `all-MiniLM-L6-v2`
> > Thanks for pointing this out. Since the frozen LM is a only pluggable component without any tuning requirements, we just follow the prior work and equip this LM to all baselines to obtain the text embeddings for fair comparisons.
> > We can easily change the embedding model to BGE or Qwen3 according to user requirements and the specific scenarios with no more than 5 lines of code change.

---

> > > ### Comment · Reviewer_oGW9 · 2025-11-26
> > >
> > > I thank the authors for the detailed rebuttal and additional experiments.
> > >
> > > The rebuttal successfully addresses the majority of my critical concerns. The new ablation studies comparing standard clustering baselines (Louvain/Leiden) effectively validate the algorithmic contribution of the "Dually-Perceived" approach. Furthermore, the clarifications regarding the "systematic seed schema construction," the "chunk preservation" mechanism for recall, and the confirmation of consistent hyperparameter usage across datasets successfully mitigate my concerns regarding the system's brittleness and contribution attribution. I hope the authors incorporate the additional results into the final version of this paper to strengthen their contribution.
> > >
> > > However, I remain unconvinced by the response regarding the Frozen LM (`all-MiniLM-L6-v2`).
> > >
> > > 1. **Scientific Validity vs. Engineering Convenience:** While "fair comparison with baselines" is a valid engineering defense, it sidesteps the scientific question. The concern persists that the complex "dual-perception" mechanism might simply be over-compensating for the insufficient expressive power of a legacy embedding model.
> > >
> > > 2. **Missing Upper Bound:** Without testing on modern, strong embeddings (e.g., `BGE-M3`), it is impossible to know the true ceiling of the framework. It remains an open question whether the proposed complex clustering is as necessary when semantic representations are inherently higher quality.
> > >
> > > 3. **Sparse Graph Fragility:** The rebuttal also failed to address the specific concern regarding the fragility of Jaccard similarity in sparse text graphs.
> > >
> > >
> > > In conclusion, the additional data has significantly strengthened the paper. **I am raising my score to Accept**, but I invite the authors to provide further discussion regarding the limitations of the embedding model and the potential redundancy of complex clustering under strong semantic encoders.

---

> ### Author Response · Authors · 2025-11-27
> **Grateful Thanks to Your Strong Support! and Further Experiments**
>
> Dear Reviewer oGW9,
>
> We are truly excited to be further acknowledged by your strong support and high recognition! Your expertise and deep technical insights has helped us substantially improve the clarity of contributions. Following your suggestions, we report the additional experiments and expanded our analysis to further address your remaining concerns.
>
> We sincerely invite you to review the experiments with BGE-M3. Actually, since MiniLM is restricted to ENG, we have already employed BGE-M3 in the real industrial CHS scenarios, and thus provides a clean upper bound for semantic representations.
>
> > **Ablation studies**
>
> We have conducted two sets of comparisons with the original UniGraphRAG: $(i)$ `all-MiniLM-v2 w/o Community Detection` and `all-MiniLM-v2 w/ Community Detection`; $(ii)$ `BGE-M3 w/o community detection`, `BGE-M3 w/ community detection`.
>
> | Variant    | HotpotQA | 2Wiki | MuSiQue | GraphRAG-Bench | Anony-CHS | Anony-ENG |
> |------|:--------:|:-----:|:-------:|:---------:|:---------:|:----------:|
> | all-MiniLM-v2   w/o Community Detection            | 78.40    | 72.30 | 42.10   | 81.65           | 39.97     | 40.59    |
> | all-MiniLM-v2  w/  Community Detection    | 81.20| **77.60**| **47.50**| **86.54**     | 42.88 | **43.26**  |
> | BGE-M3 + w/o Community Detection             | ***82.90***    | 77.20 | 44.60   | 85.27    |  ***43.46***    |    42.03    |
> | BGE-M3 + w/ Community Detection   | **85.10**| **79.40**| **49.10**| **89.10**     | **45.93** | **46.83**  |
>
> We made following observations that:
> - directly replacing MiniLM with BGE consistently boosts overall performance with approximately 3%~4% improvements. However, the necessity of community detection still remains across all datasets.
> - even with BGE, removing clustering results in a consistent drop on very most of the datasets compared with `all-MiniLM-v2  w/ Community Detection`, except for a simpler dataset HopotQA and a CHS dataset AnonyRAG-CHS. This showcases the superioty of BGE.
> - This empirically shows that dual-perception clustering is not merely compensating for weak encoders, but rather complements them by enforcing structural coherence that semantic similarity alone cannot guarantee, especially when our community detection algorithm is efficient.
>
> > **Sparse Graph Fragility**
>
> Thanks for raising this important discussion again. We value this opportunity to further demonstrate our advantages considering 'dual-perception'. We agree that if there is only pure Jaccard similarity, it will be fragile in extremely sparse or high-dimensional text graphs. Compared with Louvain and Leiden who only considers structural information, we have **two** designs to ensure the robustness:
> - **Our core intention is exactly to alleviate the fragility problem by combining 'subgraph semantics' and 'topological connections', thereby increasing the robustness to sparse graphs.**
> - The **`struct_weight$ = 0.3`** (also shown in our code `utils/tree_comm.py`), is designed to control the proper noise from structual Jaccard value while laying more weights to semantics.
>
> | Variant| Semantics | Structure |
> | :----- | :--------- | :--------- |
> | Louvain    |    ✗      |    ✓ (Edge connectivity)   |
> | Leiden |     ✗     |   ✓ (Edge connectivity)    |
> | K-Means  |    ✓ (Entity Embedding)    |    ✗  |
> | Our Dual-perception  |  ✓ (0.7 from Triple-level aggregated embedding) | ✓ (0.3 from Jaccard edges) |
>
> **Since the structural view captures multi-hop co-occurrence and neighborhood consistency and the semantic view with triple-level embedding aggregates $(entity, relation, entity/attribute)$ semantics, providing dense and stable signals even when surface-level overlap is minimal. Our weighted combination stabilizes clustering on sparse or long-tail texts.**
>
> We hope the new responses could address your remaining concerns.
>
> Thanks again for your insightful feedback!

---

### Official Review · Reviewer_128g · 2025-11-02

**Soundness:** 3
**Presentation:** 3
**Contribution:** 3
**Rating:** 4
**Confidence:** 3

**Summary:**

This paper focuses on the problem of RAG on graphs. Specifically, it concerns the problem when a set of documents are converted into a KG prior to performing RAG. Previous methods typically operate in two stages: first they construct the KG and second they can perform RAG on it. The authors argue that the current construction methods are suboptimal for the task of RAG, as RAG may require a different construction process not captured by more traditional KG construction methods. To this point, the authors propose UniGraphRAG, which attempts to unify the construction and retrieval components of GraphRAG. To do this, the authors introduce a graph construction process that is guided by a specific "seed schema", allowing for a more targeted final graph. They further use an agent to expand on the graph, by choosing relevant schema elements that are sufficiently similar. They then summarize the knowledge into several communities in a tree structure. They consider 4 types of summaries: communties (i.e., structural), keywords, triples, and attributes (in that order from top to bottom). The communities themselves are created via structural and semantic similarity measures. Retrieval is then done on the final graph, where they further use reflection to refine results. The authors include results across a variety of benchmark datasets.

**Strengths:**

1. I like the motivation, they make a strong point that the construction and retrieval process should ideally be coupled. Because in reality, it's possible that current construction methods may actually harm retrieval.

2. I like their idea for forming communities based on both structural (i.e., where in the graph) and semantic similarity. It's a subtle but good observation, as the structure is really just one piece of information that describes each node. For example, it's possible that two nodes share a high semantic similarity, but due to chance, are not near each other in the documents. As such, they don't appear in any triplets and may have low structural similarity. So this is a welcome idea.


3. The overall results are quite good. They also include results on a variety of benchmarks which is good to see.

**Weaknesses:**

1. It's unclear how the seed shemas were created. The authors only mention that they were made using queries from 2Wiki and MuSiQue (line 863). How was this done? Why is this a good strategy? My main concern is that there is no guidance for how to construct these seed schemas, in terms of their content, size, and other parameters. Furthermore, I imagine that the specific datasets may also play a big role. As such, much more information needs to be given here.

2. A separate, but bigger, concern regarding the seed schemas is that they greatly constrain the resulting graph. This is because that schema guides how the graph is created. I know that the authors introduce an agent to expand on this knowledge, however from my understanding, it should still only really consider related information. However, I'd argue that this is suboptimal in real-world applications for two reasons: **(1)** It's unpredictable what kind of query a user may ask. As such, the constructed graph may have been optimized for queries that some user doesn't ask. In essence, the problem is capturing the long tail of user queries, which I'm unsure this method can handle. **(2)** For multi-hop questions, it is often necessary to take "detours" through pieces of information that may not seem relevant. However, by restricting the content in the graph, we are effectively closing off these paths, making it harder to solve these queries. This goes back to my last point, in that while this may be okay for many queries, it could very well hurt the long tail of user queries that are more complex or are on the margins of the seed schema.

3. I don't see why having a more efficient graph construction process is very important, as this is a simply a pre-processing step and only needs to be done once. As such, the benefit is marginal at best.

4. It's unclear to me how the retrieval and construction process are actually "unified". At the end of the day, they are still separate processes. Maybe the proposed construction process is better for retrieval, but it nonetheless is it's own thing. What I mean, is for them to really be unified, the construction process would have to be conditional on the retrieval which it isn't (and would be prohibitvely expensive). This is a minor weakness, but I think it would be better for the authors to be more process in their language.

**Questions:**

1. I'd appreciate your perspective on my concerns in weakness 2, regarding the limitations of constructing the graph via a seed schema. In particular, I'm concerned that it may omit some information that is important for a subset of queries. [**Note**: This is my main concern and I'd be happy to raise my score depending on the answer]

2.  Can you give more details about the seed schemas used (see weakness 1).

---

> ### Author Response · Authors · 2025-11-22
> **Responses to Reviewer 128g (1/3)**
>
> Dear Reviewer 128g,
>
> We would like to express our sincere gratitude for your interest and positive assessment of our paper, especially on motivation and community detection. Receiving such insightful comments and recognition from community experts is truly inspiring.
>
> We also wish to invite the reviewer to check our rebuttal and all the dscussions will be correspondingly revised in the updated version following your constrcutive suggestions.
>
> **Responses to weaknesses and questions**
> ---
> >**W1&Q2: Details about seed schema curation.**
>
> Thanks for pointing this out. In our submission with anonymous code, we have shown all the schemas we used under folder `/schemas`. Below we provide a clear, systematic explanation of how we create seed schemas, why this strategy is principled, and how users should determine the content, size, and parameters when applying UniGraphRAG to new domains. Following your comment, the details will be added in the revised version for clearer illustration.
>
> - **Systematic and label-agnostic seed schema construction**:
> We would like to make further explanations about how we prepare the seed schema in a systematic way. This procedure requires no dataset labels and is fully reproducible from any raw corpora.
> (i) For structured benchmarks like GraphRAG-Bench, we directly collect the 'Table of Content' or chapter headers as the high-quality potential topics, then feed them to LLMs and extract the high-level seed schema set for this domain. This will help us quickly grasp the domain knowledge hierarchy for schema preparation.
> (ii) While for unstrcutured datasets with no explicitly clear hierarchy like HotpotQA and our proposed AnonyRAG-CHS/ENG, We compute n-gram statistics (n=2,3,4) over the corpus to surface the top 200 high-frequency candidates and employ the LLMs to generate a core schema based on them, which is also extremely efficient, e.g., Person / Organization / Location / Event. Experts are then required to make slight pruning to ensure the conciseness.
> - **Why seed schema is a good strategy?**
> Considering our schema, we would like to clarify that this is a `distantly supervised` knowledge recognition that ensures both quality and coverage. The extracted triple does not have to be $(h_{targetted}, r_{targetted}, t_{targetted})$. Once there is a head/tail/relation matches the schema, LLMs will be encouraged to extract the related knowledge, which will greatly ensure the scalable coverage for detailed and long-tail knowledge.
> For instance, we can have:
> $(h_{targetted}, r_{new}, t_{new})$,
> $(h_{targetted}, r_{targetted}, t_{new})$,
> $(h_{new}, r_{targetted}, t_{new})$, etc.
> The newly identified knowledge type will then be considered whether to be included to enrich the seed bank. Therefore, our seed schema only anchors types, not specific facts, making a good balance between valuable content and noisy information. Moreover, the extraction agent is not constrained to extract the triples where the entire (h,r,t) strictly match the schema. The schema provides a good starting search space in a distantly supervised way, and the agent is explicitly designed to discover and propose new entity types/relations/attribute types to continuously enrich the seed bank.
> - **Schema size, content, and parameters**
> Across all benchmarks and domains, our schemas follow a consistent scale:
> Entity types: 5～20
> Relations: 5～20
> Attribute types: 5～20
> These ranges are large enough to cover domain abstractions, yet small enough to avoid noise and overfitting. We also allow adaptive expansion (our agent handles this automatically), so seed schemas remain compact.
> oOur actual schemas are listed under the /schemas folder in our anonymous code, and the revised paper will include examples and guidelines.
>
> A clarification for Line 863 here is that this is actually the way how we create QA pairs for AnonyRAG-CHS/ENG, not for schemas. We learn from the existing patterns in multi-hop QA datasets to prepare high-quality anonymous QA datasets. We will revise this paragraph in the updated submission to make it clearer.
>
> Here, we attach a sample schema for GraphRAG-Bench:
> ```
> {
>     "Nodes": [
>         "concept",
>         "prerequisite",
>         "formula",
>         "theorem",
>         "method",
>         "diagram",
>         "equation",
>         "topic"
>     ],
>     "Relations": [
>         "requires",
>         "is_part_of",
>         "contains",
>         "illustrates",
>         "explains",
>         "proves",
>         "applies_to",
>         "uses",
>         "is_example_of",
>         "solves",
>         "is_defined_in",
>         "references",
>         "builds_upon"
>     ],
>     "Attributes": [
>         "title",
>         "name",
>         "description",
>         "expression",
>         "definition",
>         "example",
>         "instructions",
>         "exercise"
>     ]
> }
> ```

---

> > ### Author Response · Authors · 2025-11-22
> > **Responses to Reviewer 128g (2/3)**
> >
> > > **W2 & Q1: Graph quality constrained by schema.**
> >
> > Thanks for raising this thoughtful concern. We value this good opportunity to showcase the superiorty of our schema-enhanced graph construction in a distantly supervised manner and how we preserve the probable long-tail knowledge that could be used.We address your concerns below from both a design principle perspective and an empirical perspective, and we will incorporate these clarifications into the revised paper.
> >
> > - **Existing pipelines**
> > Let's first take a look at two existing pipelines of graph construction.
> > (i) Purely open extraction adopted by Microsoft GraphRAG and LightRAG. Though they could extract as much information as possible that may also cover the long-tail knowledge, a lot of noise will be inevitably introduced. This makes the graph much bigger and therefore, most importantly, harm the downstream retrieval performance. (ii) OpenIE+LLM. HippoRAG 1&2 and GFM-RAG adopt OpenIE, which is a much stricter way that recognize entities based on rules and traditional NLP techniques. This is indeed an attempt to constrain the extraction for more valuable content but lack of scalability to handle domain-specific scenarios.
> > - **Schema ≠ Limits**
> > Repeated as the aforementioned, Considering our schema, we would like to clarify that this is a `distantly supervised` knowledge recognition that ensures both quality and coverage. The extracted triple does not have to be $(h_{targetted}, r_{targetted}, t_{targetted})$. Once there is a head/tail/relation matches the schema, LLMs will be encouraged to extract the related knowledge, which will greatly ensure the scalable coverage for detailed and long-tail knowledge.
> > For instance, we can have
> > $(h_{targetted}, r_{new}, t_{new})$,
> > $(h_{targetted}, r_{targetted}, t_{new})$,
> > $(h_{new}, r_{targetted}, t_{new})$, etc.
> > The newly identified knowledge type will then be considered whether to be included to enrich the seed bank.
> > Therefore, our seed schema only anchors types, not specific facts, making a good balance between valuable content and noisy information. Moreover, the extraction agent is not constrained to extract the triples where the entire (h,r,t) strictly match the schema. The schema provides a good starting search space in a distantly supervised way, and the agent is explicitly designed to discover and propose new entity types/relations/attribute types to continuously enrich the seed bank.
> > A simple expanded case here:
> > ```
> > {
> > 'Nodes': ['Concepts', 'Theorem'(expanded)],
> > 'Relations': ['Subtopic_of', 'Applies_to'(expanded)],
> > 'Attributes': ['Definitions', 'Applications'(expanded)]
> > }
> > ```
> > Based on this, the expanded relations will naturally enable fluent multi-hop reasoning (detours) through the paths like `subtopic_of` and the expanded `prerequisite_of`.
> > - **Original chunks never discard**
> > We indeed have also maintained a bi-directional linking between the extracted triples and the chunk to preserve the original source. In our released anonymous code, we use `nanoid` to represent each chunk and store it in the graph structure, column 'properties'. This could greatly improve the performance handling questions about 'long-tail knowledge' by preserving the original chunks that contain the source texts.
> > - **Experimental evidence**
> > To further address your concerns and verify this design, we would like to report the experimental results (Reject Acc. for three multi-hop QA datasets and Open Acc. for GraphRAG-Bench) based on DeepSeek V3 using different graphs generated by different methods while keeping the same UniGraphRAG retriever (with no type filtering/schema-based decomposition in retrieval). The results showcase the superioty of our constructed graph (knowledge tree) based on the schema, especially on MuSiQue and 2Wiki, where multi-hop detours are essential.
> >
> > | [Graph]+UniGraphRAG | HotpotQA | 2Wiki | MuSiQue | G-Bench |
> > | :--- | :---------: | :---------: | :---------: | :---------: |
> > | [GraphRAG] | 59.60 | 39.40 | 22.80 | 75.64 |
> > | [LightRAG]| 61.20 | 41.10 | 31.20 | 79.56 |
> > | [HippoRAG] | 77.30 | 74.00 | 43.20 | 80.55 |
> > | **UniGraphRAG**| **81.20** | **77.60** | **47.50** | **86.54** |
> >
> > We hope this detailed explanation addresses your concerns, and thank you again for raising this important point.

---

> > > ### Author Response · Authors · 2025-11-22
> > > **Responses to Reviewer 128g (3/3)**
> > >
> > > > **W3. Why care about efficient graph construction?**
> > >
> > > Thank you for raising this point. While graph construction is indeed a pre-processing step, in real industrial RAG it is not a “build once and forget” stage. This paper is motivated by real industrial needs and we would like to clarify why efficiency is a critical factor in downstream deployments.
> > > - **The size of commercial corpora could be extremely big.**
> > > Even one-time graph construction could be expensive and time-consuming when commercial settings index millions of docs across verticals;
> > > - **Graphs change frequently.**
> > > Many real-world applications update graphs daily or hourly (personal knowledge bases (KBs), financial KBs, HR KBs with candidate CVs, etc.). Each time when the KB is updated, we need to incrementally update the graph that naturally requires an efficient one. Both real-world situations urge the efficient graph construction.
> > > - We agree in particular scenarios with fixed corpus, efficiency provides a smaller benefit, but it is still truly impactful and practical in the very most of industries and real-world applications.
> > >
> > > > **W4. Clarifying the use of the term 'Unified'**
> > >
> > > Thank you for highlighting this. We agree that the term 'unified' for retrieval-conditioned construction, is not the 'unified' what we intend to claim. We refer 'unified' as a shared knowledge space during both extraction and retrieval, rather than a unified optimization loop.
> > > While existing methods suffer from extraction and retrieval being inconsistent, our unified framework prevents the typical mismatch between construction-time assumptions and retrieval-time reasoning.
> > >
> > > We appreciate your comment, in the revision, we will further define the idea with clearer demonstrations.

---

> > > > ### Comment · Reviewer_128g · 2025-11-25
> > > >
> > > > I appreciate the detailed response.
> > > >
> > > > You've addressed some of my concerns with regards to the seed schemas. I am still somewhat concerned that they be limiting in certain contexts, but I nonetheless understand your argument.
> > > >
> > > > I also think editing the paper with a clarification on "unified" would be beneficial and help avoid any misunderstandings.
> > > >
> > > > I've raised my score to a 6.

---

> > > > > ### Author Response · Authors · 2025-11-26
> > > > >
> > > > > Dear Reviewer 128g,
> > > > >
> > > > > Thank you so much for your support and thoughtful follow-up! We truly appreciate your open-minded engagement with our clarification on seed schemas.
> > > > >
> > > > > We fully understand your remaining concerns regarding the potential limitations of seed schemas in particular contexts. In the revised version, we will further clarify the intended flexibility of our schema design, the mechanisms for adaptive expansion, and how UniGraphRAG avoids enforcing constraints in practice. We will also revise the introduction and methodology sections to explicitly explain our motivation and the conceptual role of 'unified' to avoid any possible misunderstanding.
> > > > >
> > > > > Thank you again for your constructive feedback and for helping us strengthen the clarity and presentation of the work. We sincerely appreciate your time and consideration.

---

### Author Response · Authors · 2025-11-28
**General Responses and Summary During This Special Period**

Dear AC and Reviewers,

We would like to take this unique opportunity to sincerely appreciate all of your efforts and insightful comments with a summary. It is very encouraging to be uniformly and highly recognised by all the reviewers in terms of our motivation and novelty on (i) being unified with schema and (ii) dual-perception community detection that could push GraphRAG a step forward in real-world applications. The discussions have truly improved our paper to better clarify our contributions and novelty.

**With the score raising from from 6/4/4/2 to `8/6/4/4`, we are also deeply encouraged since our rebuttal has successfully addressed the concerns from most of the reviewers before the system bug was revealed** (though it's a great pity that there are still ongoing/unstarted discussions between several reviewers since the system is locked now).

> Reviewer Scores, Discussion Status, and Addressed Concerns

| Reviewer | Original → Updated Score | Discussion Status | Key Acknowledgements / Remaining Concerns |
|---------|---------------------------|-------------------|--------------------------------------------|
| **oGW9** | **`6 → 8`** | Ongoing | 1. Confirmed effectiveness of added ablations; 2. appreciates strengthened contributions; 3. requests brief clarification on embedding-model limitations & necessity of complex clustering (we addressed but the discussion was suspended due to system issue). |
| **128g** | **`4 → 6`** | Done | 1. Schema curation concerns addressed; 2. acknowledges clarification on “unified” design and request this demonstration in the paper; 3. satisfied with supplemented experiments. |
| **rQJp** | **`2 → 4`** | Ongoing | 1. Recognizes reply; 2. acknowledges added experiments and explanations;  3. still finds system complex (we addressed by proving no tuning on the hyperparameters but the discussion was suspended due to system issue). |
| **vYRA** | 4 → 4 | Not started | No discussion due to system issue. |

In conclusion, we would like to sincerely invite AC to check the discussion history and the addressed main concerns include:

| Concern Category | Evidence Provided | Reviewers Confirming the Resolution |
|------------------|------------------|------------------------------------|
| **Schema curation & manual efforts** | Demonstrations + clarification of unified schema design |  **oGW9 (8)**, **128g (6)**, **rQJp(4)** |
| **Supplementary ablation studies** | **NINE** additional experiments verifying robustness and design choices | **oGW9 (8)**, **128g (6)** |
| **System complexity & hyperparameter tuning** | Code + demonstrations showing generalizability using identical hyperparameters across datasets | **oGW9 (8)** |

Though the discussions between Reviewer vYRA (Score: 4) has not started yet and ones between Reviewer oGW9 (Score: 8, ongoing) and Reviewer rQJp (Score: 4, ongoing) are still in the half way with new experiments due to this system issue, we still would like to sincerely thank all the reviewers for your insightful, professional and construction reviews!

These are the best gifts we received during this conference submission, and we hope this summary will quickly assist AC in grasping the acknowledgements we have received and the concerns we have addressed during the rebuttal.

---

### Meta-Review · Area_Chair_xXoo · 2026-01-06

**Summary:**

This paper proposes UniGraphRAG, a vertically unified agentic framework for graph retrieval-augmented generation that aligns graph construction, indexing, and retrieval through a shared schema. On the construction side, a schema-bounded extraction agent produces entity, relation, and attribute triples while allowing controlled schema expansion to handle domain shift. On the indexing and retrieval side, the method builds a four-level knowledge tree using dual-perception community detection that combines graph topology and semantic similarity, and a schema-aware retriever that decomposes complex queries with iterative reflection.The paper also introduces bilingual anonymous benchmarks and an Anonymity Reversion evaluation to reduce the impact of LLM memorization when assessing GraphRAG systems.

Reviewers’ key concerns centered on the perceived dependence on seed-schema design, potential brittleness or recall loss under long-tail queries, unclear attribution across multiple coupled components, missing or unfair baseline comparisons, and sensitivity to embedding choices and structural heuristics. After carefully reading the entire submission and the rebuttal materials (noting that discussion was partially curtailed by the platform lock), including the added ablations and baseline studies,  the AC is convinced the authors addressed the major issues and that the unified schema plus hierarchical indexing yields a robust improvement, so acceptance is recommended.

**Reviewer Concerns:**

The rebuttal substantially addressed concerns about manual schema curation by providing a reproducible seed-schema construction procedure, practical size guidelines, and clarifying that the schema anchors types rather than restricting facts, with automatic expansion and chunk anchoring to preserve recall. It also addressed requests for clearer contribution attribution by adding detailed ablations across schema, community detection, decomposition, and reflection, and by benchmarking the proposed clustering against standard alternatives.
Concerns about missing comparisons and agentic bias were partially resolved through added experiments that augment baselines with standard query decomposition and reflection and through comparisons to agentic RAG variants, alongside fixes to presentation issues such as code link formatting and figure typos. Remaining outstanding issues are mainly about real-world adoption risk: the pipeline is still complex to implement and audit end-to-end, and while stronger-embedding experiments suggest the approach generalizes, the community would benefit from the final paper more explicitly stating when a seed schema might fail to cover truly open-ended query distributions and what safeguards or diagnostics to use.

**Reviewer Scores:**

Reviewer oGW9 already moved from 6 to 8 after seeing expanded ablations and would likely have stayed at 8 (or edged higher) had the discussion continued through the final embedding and sparsity analyses.Reviewer 128g raised from 4 to 6 after the clarifications on seed schemas and “unified” framing, and with full discussion time would likely remain at 6 with a small chance of a 7 if the final revision further emphasizes long-tail coverage and limits. Reviewer rQJp increased from 2 to 4 but still expressed skepticism about complexity and tuning, and I expect a fuller discussion incorporating the new ablations, fixed hyperparameter defaults, and clearer retriever specification could have brought them to around 5. Reviewer vYRA could not participate in discussion; given the added comparisons to agentic baselines, responses on hyperparameters and anonymization analysis, and corrections to presentation issues, I believe they would likely have increased from 4 to about 5.

---

### Decision · Program_Chairs · 2026-01-26

Accept (Poster)